# LATEC – A benchmark for large-scale attribution & attention evaluation in computer vision

## Abstract

Explainable AI (XAI) is a rapidly growing domain with a myriad of proposed methods as well as metrics aiming to evaluate their efficacy. However, current literature is often of limited scope, examining only a handful of XAI methods and employing one or a few metrics. Furthermore, pivotal factors for performance, such as the underlying architecture or the nature of input data, remain largely unexplored. This lack of comprehensive analysis hinders the ability to make generalized and robust conclusions about XAI performance, which is crucial for directing scientific progress but also for trustworthy real-world application of XAI. In response, we introduce LATEC, a large-scale benchmark that critically evaluates 17 prominent XAI methods using 20 distinct metrics. Our benchmark systematically incorporates vital elements like varied architectures and diverse input types, resulting in 7,560 examined combinations. Using this benchmark, we derive empirically grounded insights into areas of current debate, such as the impact of Transformer architectures and a comparative analysis of traditional attribution methods against novel attention mechanisms. To further solidify LATEC's position as a pivotal resource for future XAI research, all auxiliary data—from trained model weights to over 326k saliency maps and 378k metric scores—are made publicly available. The benchmark is hosted at: `https://github.com/kjdhfg/LATEC`

## 1 Introduction

Explainable AI (XAI) methods have become essential tools in numerous domains, allowing for a better understanding of complex machine learning decisions. The most prevalent XAI methods originate from the domain of saliency maps Simonyan et al. (2013). For a systematic review of other XAI methods such as counterfactual examples (Wachter et al., 2018) or concept testing Kim et al. (2018) we refer to Speith (2022). As the diversity and abundance of proposed saliency XAI methods expand alongside their growing popularity, ensuring their reliability becomes paramount (Adebayo et al., 2018). Given that there is no clear "ground truth" for individual explanations (as e.g. discussed in Adebayo et al. (2020)), the trustworthiness of XAI methods is typically determined by examining three key aspects: their accuracy in reflecting a model's reasoning ("faithfulness") (Bach et al., 2015; Samek et al., 2017), their stability under small changes ("robustness") (Yeh et al., 2019; Alvarez Melis & Jaakkola, 2018), and the understandability of their explanations ("complexity") (Chalasani et al., 2020; Bhatt et al., 2021). Beyond qualitative assessments such as in Doshi-Velez & Kim (2017); Ribeiro et al. (2016); Shrikumar et al. (2017), which can be influenced by human biases and don't always scale well (as shown by Wang et al. (2019); Rosenfeld (2021)), a wide array of metrics have been introduced to quantitatively evaluate XAI methods based on these key aspects.

However, the current state of validation in XAI research exhibits two major shortcomings. First, many studies restrict their analyses to a limited set of methods and evaluation metrics. Second, current research often fails to consider how different underlying model architectures (Yona & Greenfeld, 2021) and varied input modalities (Budding et al., 2021) directly impact the performance of XAI methods. As a consequence, our current understanding of XAI performance and its generalizability is limited, making it challenging for practitioners to determine a reliable XAI method for their specific use case. Table 1 demonstrates this fragmented landscape specifically for the domain of computer vision, including discrepancies found across studies, with some methods, such as Grad-

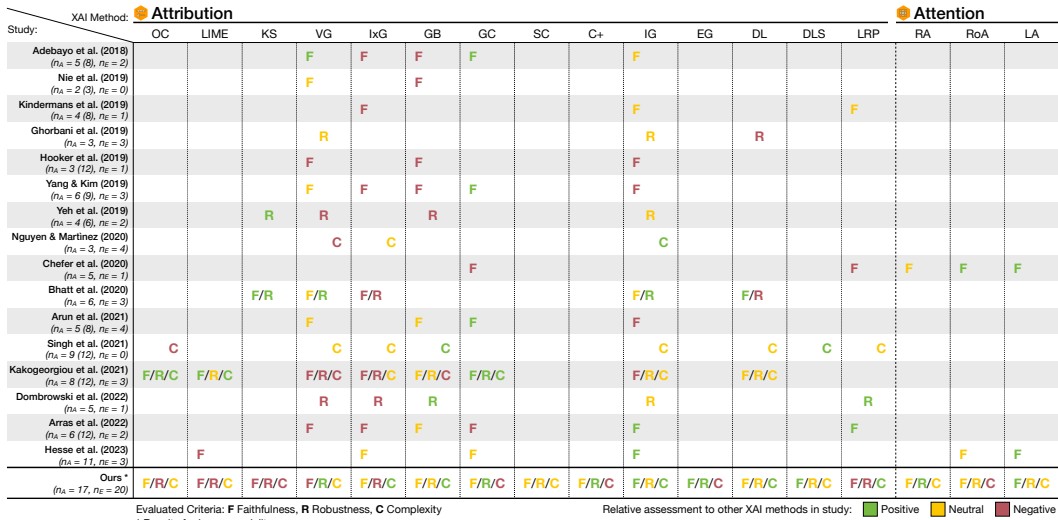

| Study: | OC | LIME | KS | VG | IxG | GB | GC | SC | C+ | IG | EG | DL | DLS | LRP | RA | RoA | LA |
|---|---|---|---|---|---|---|---|---|---|---|---|---|---|---|---|---|---|
| Adebayo et al. (2018) ($n_A$ = 5 (8), $n_E$ = 2) | | | | F | F | F | F | | | F | | | | | | | |
| Nie et al. (2019) ($n_A$ = 2 (3), $n_E$ = 0) | | | | F | | F | | | | | | | | | | | |
| Kindermans et al. (2019) ($n_A$ = 4 (8), $n_E$ = 1) | | | | | F | | | | | F | | | | F | | | |
| Ghorbani et al. (2019) ($n_A$ = 3, $n_E$ = 3) | | | | R | | | | | | R | | R | | | | | |
| Hooker et al. (2019) ($n_A$ = 3 (12), $n_E$ = 1) | | | | F | | F | | | | F | | | | | | | |
| Yang & Kim (2019) ($n_A$ = 6 (9), $n_E$ = 3) | | | | F | F | F | F | | | F | | | | | | | |
| Yeh et al. (2019) ($n_A$ = 4 (6), $n_E$ = 2) | | | R | R | | R | | | | R | | | | | | | |
| Nguyen & Martinez (2020) ($n_A$ = 3, $n_E$ = 4) | | | C | C | | | | | | C | | | | | | | |
| Chefer et al. (2020) ($n_A$ = 5, $n_E$ = 1) | | | | | | | F | | | | | | | F | F | F | F |
| Bhatt et al. (2020) ($n_A$ = 6, $n_E$ = 3) | | F/R | F/R | F/R | | | | | | F/R | | F/R | | | | | |
| Arun et al. (2021) ($n_A$ = 5 (8), $n_E$ = 4) | | | | F | | F | F | | | F | | | | | | | |
| Singh et al. (2021) ($n_A$ = 9 (12), $n_E$ = 0) | C | | | C | C | C | | | | C | | C | C | C | | | |
| Kakogeorgiou et al. (2021) ($n_A$ = 8 (12), $n_E$ = 3) | F/R/C | F/R/C | | F/R/C | F/R/C | F/R/C | F/R/C | | | F/R/C | | F/R/C | | | | | |
| Dombrowski et al. (2022) ($n_A$ = 5, $n_E$ = 1) | | | | R | R | R | | | | R | | | | R | | | |
| Arras et al. (2022) ($n_A$ = 6 (12), $n_E$ = 2) | | | | F | F | F | F | | | F | | | | F | | | |
| Hesse et al. (2023) ($n_A$ = 11, $n_E$ = 3) | | F | | F | | F | | | | F | | | | | | F | F |
| Ours * ($n_A$ = 17, $n_E$ = 20) | F/R/C | F/R/C | F/R/C | F/R/C | F/R/C | F/R/C | F/R/C | F/R/C | F/R/C | F/R/C | F/R/C | F/R/C | F/R/C | F/R/C | F/R/C | F/R/C | F/R/C |

Evaluated Criteria: **F** Faithfulness, **R** Robustness, **C** Complexity
\* Results for image modality
Relative assessment to other XAI methods in study: ■ Positive ■ Neutral ■ Negative

Table 1: Showing gaps and inconsistencies between 16 relevant related studies evaluating XAI methods. Colors coincide with their ranking inside the study depending on the evaluation criteria. $n_A$: Amount of XAI methods, number including slightly adapted versions in parenthesis as we do not deem, e.g. IG and Smooth-IG, as two significantly different methods. $n_E$: Number of evaluation metrics. If $n_E = 0$ the study was conducted either qualitatively or experiment-based without scores. The color indicating the relative performance of our work is always based on the image modality.

CAM, receiving contradictory assessments depending on the evaluation setup Chefer et al. (2021); Arras et al. (2022); Adebayo et al. (2018); Yang & Kim (2019); Arun et al. (2021).

Further, the present shortcomings in XAI research validation prevent us from answering a multitude of general questions pivotal to the field's advancement. For instance:

Q1: How does the performance of attention versus attribution methods differ in practice?
Q2: Does the efficacy of XAI methods vary across different computer vision modalities?
Q3: With the ascendency of Transformer architectures Dosovitskiy et al. (2021), is there a potential misalignment with established attribution-based XAI methods?

These questions help direct research to make XAI methods more practical and relevant for the changing world of AI models and applications. In response to these challenges, we present LATEC, the first comprehensive benchmark tailored for a holistic evaluation in the field of XAI in computer vision. LATEC encompasses 17 of the most widely-used saliency XAI methods, including attention-based methods, and evaluates them using 20 distinct metrics (see Figure 1). Notably, LATEC integrates a variety of model architectures, and, to extend the evaluation spectrum beyond traditional 2D images, we included point cloud and volume data, adapting XAI methods as necessary to suit these modalities. In total, LATEC assesses 7,560 unique combinations.

The presented benchmark offers a practical platform for the community to systematically investigate general questions in the field of XAI. As proof of this utility, we provide a detailed empirical analysis based on the three questions Q1-Q3. Moreover, in support of future research, we've made all intermediate data, including 326,790 saliency maps and 378,000 evaluation scores, publicly accessible. This "LATEC dataset" is aimed at simplifying the benchmarking process for new methods and metrics in XAI.

## 2    THE LATEC BENCHMARK

The LATEC benchmark includes a framework and dataset (see Figure 1). The framework structures experiments in six stages for diverse large-scale studies, while the LATEC dataset provides reference data for evaluating new XAI methods or exploratory analysis.

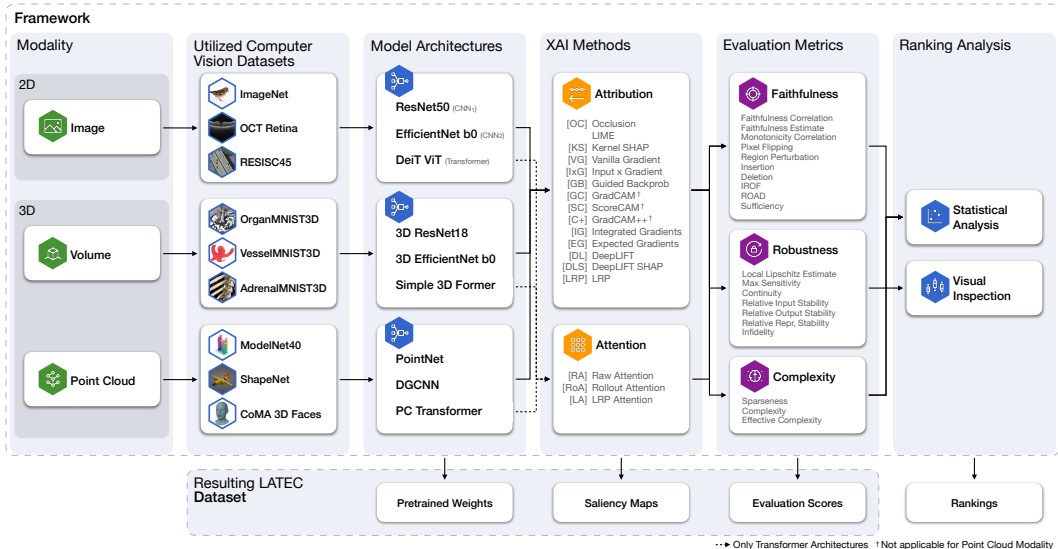

Figure 1: Structure of the benchmark with the output data of each stage. Colors reflect the stages.

## 2.1 THE FRAMEWORK: DESIGNING DIVERSE LARGE-SCALE EVALUATIONS

**Utilized computer vision datasets**    For the image modality we use ImageNet (IMN) (Deng et al., 2009), UCSD OCT retina (OCT) (Kermany et al., 2018) and RESISC45 (R45) (Cheng et al., 2017), for the volume modality the Adrenal-(AMN), Organ-(OMN) and VesselMedMNIST3D (VMN) datatsets (Yang et al., 2023), and for the point cloud modality the CoMA (CMA) (Ranjan et al., 2018), ModelNet40 (M40) (Wu et al., 2014) and ShapeNet (SHN) (Chang et al., 2015) datasets.

**DL architectures**    On each utilized computer vision dataset except IMN, where we take pretrained models, we train three models to achieve the architecture-dependent SOTA performance on the designated test set (if available, see Appendix B for a detailed description of all model trainings and hyperparameters). For the image modality, we use the ResNet50, EfficientNetb0, and DeIT ViT (Touvron et al., 2022) architectures, for the volume modality the 3D ResNet18, 3D EfficientNetb0, and Simple3DFormer (Wang et al., 2022) architectures, and for the point cloud modality the Point-Net, DGCNN and PC Transformer (Guo et al., 2021) architectures. The first two architectures are always CNNs ($CNN_1$ and $CNN_2$), and the third is a Transformer.

**XAI methods**    In total, we include 17 XAI methods, 14 attribution methods: Occlusion (OC) (Zeiler & Fergus, 2013), LIME (on feature masks) (Ribeiro et al., 2016), Kernel SHAP (KS, on feature masks) (Lundberg & Lee, 2017), Vanilla Gradient (VG) (Simonyan et al., 2013), Input x Gradient (IxG) (Shrikumar et al., 2017), Guided Backprob (GB) (Springenberg et al., 2015), Grad-CAM (GC) (Selvaraju et al., 2017), ScoreCAM (SC) (Wang et al., 2020), GradCAM++ (C+) (Chat-topadhay et al., 2018), Integrated Gradients (IG) (Sundararajan et al., 2017), Expected Gradients (EG, also called Gradient SHAP) (Erion et al., 2020), DeepLIFT (DL) (Shrikumar et al., 2017), DeepLIFT SHAP (DLS) (Lundberg & Lee, 2017), LRP (with $\epsilon$-, $\gamma$- and $0^+$-rules depending on the model architecture) (Binder et al., 2016), and three attention methods: Raw Attention (RA) (Doso-vitskiy et al., 2021), Rollout Attention (RoA) (Abnar & Zuidema, 2020) and LRP Attention (LA) (Chefer et al., 2021). While the attribution methods are applied to all model architectures, the attention methods can only be applied to the Transformer-based architectures. For comparison reasons, we only consider the original methods without adaptations, as several other works (Hooker et al., 2019; Yang & Kim, 2019) already showed that advancing methods by VarGrad (Adebayo et al., 2018) or SmoothGrad(-squared) (Smilkov et al., 2017) can, in general, improve results. We tuned the XAI parameters per modality from the qualitative evaluation (see Figure 2) to not bias the quantitative evaluation results (see Appendix C for all XAI method parameters).

**Evaluation metrics**    The evaluation metrics are grouped into three criteria: faithfulness (is the explanation following the model behavior?), robustness (is the explanation stable?), and complexity

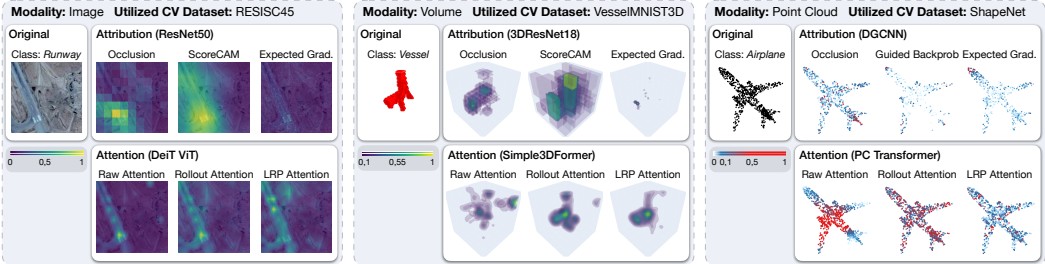

Figure 2: Illustrative saliency maps for all three modalities. The upper row shows three attributions, respectively, and the lower row, three attention-based methods. We observe how all XAI methods highlight the runway in the image and the vessel for the volume modality but with different granularity and focus. For the point cloud plane, explanations are less understandable, with attribution methods highlighting single points at the front tip, rudder, or wing tips.

(is the explanation concise and human understandable?). As every metric is again a proxy for the respective criteria, we employ several to get more reliable results. In Appendix G, we provide a detailed review of the metrics to examine their behaviour based on the criteria they approximate. We utilize in total 20 evaluation metrics, of which 10 evaluate faithfulness, seven robustness, and three complexity (see Appendix D for a detailed description of every single metric). We tune their parameters per dataset as some parameters depend on dataset properties (see Appendix subsection D.2 for all parameters).

**Ranking analysis**     As the nominal evaluation scores have no semantic meaning and their scales differ between datasets, we analyze the XAI methods based on their ranking. To this end, we compute the median evaluation score over the observations on the dataset level and rank the methods according to the metric. See Appendix E for a detailed flow chart of how we get from evaluation scores to rankings and Appendix K for the distribution of all scores before ranking. We utilize the rankings for statistical analysis in the subsequent empirical study.

## 2.2    THE LATEC DATASET: REFERENCE DATA FOR STANDARDIZED EVALUATION

The resulting data of the three stages, which comprise the LATEC dataset, include pretrained model weights (excluding IMN), saliency maps, and evaluation scores. Thanks to the LATEC dataset, future experiments can start at a certain stage and use the results from the previous stage without recomputing everything again, e.g. when testing out a new evaluation metric on the existing saliency maps, preserving comparability. For the LATEC dataset, we compute per dataset saliency maps for the entire test set or 1000 observations depending on which size is smaller (on the validation set if the test set is unavailable), from which we sample 50 observations to compute evaluation scores for all 7,560 combinations. In total, the LATEC dataset consists of 326,790 saliency maps and 378,000 evaluation scores. As for such large datasets, the size can go into the hundreds of gigabytes. To save disk space, saliency maps could be cast from 64-bit precision to 32 or even 16-bit. We would, however, strongly advise against this, as even casting to 32-bit precision introduced numerical instability in our experiments due to the rounding of attribution and attention values, resulting in all-zero saliency maps and *nan* or *inf* evaluation scores. Further, as ranking lengths between CNN and Transformer architectures differ (attention methods only for Transformer architectures), we recompute rankings in the subsequent study, which aggregate over all three architectures by first combining the normalized evaluation scores per model architecture and then computing the ranking, preserving equal length between rankings (see Appendix E).

To ensure a standardized setting with fair comparability between XAI methods over all possible experiment set-ups and aggregation levels, we take precautions regarding e.g. different types of feature attributions or the conversion of all metrics to single scores (see Appendix F for all detailed procedures). LRP requires non-negative activation outputs Montavon et al. (2019), leading us to a replacement of such activation functions (i.e. GeLU, leakyReLU) in CNN models, but we keep them for Transformer models, as they are central to the architecture and therefore also to our benchmark, and apply the $0^+$-rule instead.

### 2.3 Advancing current XAI methods and evaluation metrics for 3D data

While many XAI methods and evaluation metrics are independent of the input space dimensions, especially methods leveraging perturbations, interpolations for up- and down-scaling or segmentation are not. Our implementation builds upon the work from Kokhlikyan et al. (2020) and Hedström et al. (2023b) for XAI methods and evaluation metrics for 1D and 2D images, and we extended it to 3D volume and point cloud data. Both modalities come with their own specifies, e.g. that local neighborhoods have to be defined via k-nearest neighbors (KNN) in point cloud data and not 2D or 3D patches as in image or volume data. For the XAI methods, we advanced e.g. OC, LI, and KS by the adoption of 3D patches, all three CAM methods with 3D interpolation, all attention-based methods with 3D and KNN-based interpolations, and LA with relevance backpropagation for the Simple3DFormer and PC Transformer architectures. As the adoption of the CAM methods for point cloud data and more complex architectures than PointNet is not trivial, we deem it out of scope for this paper and do not include them in our point cloud experiments. In the case of evaluation metrics, we adapted e.g. perturbation applying metrics to 3D patches or point-based perturbations, the superpixel segmentation in IROF by 3D Slic and KMeans clustering and padded x-axis transversal for the volume and point cloud data in Continuity. Additionally, we modified all methods and metrics to function with $(x, y, z)$ volume and $(n, 3)$ point cloud dimensions. All adaptations were tested for their coherency, and illustrative saliency maps can be observed in Figure 2. We refer to Appendix C and Appendix subsection D.2 for all implementation details.

## 3 Empirical study

Table 2 provides an overview of our empirical study, with a more detailed version in Appendix H. In this study, we address three pivotal questions (Q1 - Q3) concerning XAI that remained unanswered in prior work. We analyze the data pertaining to these questions in the subsequent three subsections, with a concise presentation of our findings in subsection 3.4.

### 3.1 Q1: How does the performance of attention versus attribution methods differ in practice?

Traditionally, attribution methods are used to compute saliency maps, but since the rise of Transformers in computer vision, more methods utilizing the Transformer architecture inherent attention are explored. As researchers and engineers will be faced more and more with this choice between the two types of XAI methods, we will analyze if and how they differ in application. To quantitatively show that there are meaningful patterns between the rankings of all XAI methods in the dataset, we utilized Multidimensional Scaling (MDS) to find distance-preserving clusters across all rankings. The MDS plot in Figure 3 (a.) highlights distinct ranking similarities among analogous operating methods, especially for the attention cluster. Within the attribution methods, the subgroups of CAM methods (GC, C+, SC) and linear surrogate model methods (LI, KS), but also IG, with IxG and DL, emerge as similarly clustered. When comparing attention and attribution methods between evaluation criteria, we observe in Figure 3 (b.) a large difference in complexity and a smaller difference in robustness while the difference in faithfulness is substantially lower and insignificant. A more nuanced exploration into the attention and attribution methods (see Figure 3 (c.)) reveals already more complex relationships.

Gradient and Deep Taylor Decomposition principle-based methods, including IxG, IG, DL, and LRP, are ranked significantly less complex compared to the CAM and attention methods. In our opinion, this observation is counterintuitive when comparing the complexity rankings to the maps in Figure 2, based on which we would classify CAM and attention methods as more clearly arranged and less noisy. While all three complexity metrics (Complexity (Bhatt et al., 2020), Effective Complexity (Nguyen & Martínez, 2020) and Sparseness (Chalasani et al., 2020)) were also explicitly proposed for image data, we notice that they all treat each pixel, voxel or point independent of each other, ignoring locality and favoring methods which attribute to the smallest set of single pixels. As this approach possibly transfers to low dimensional images such as MNIST Lecun et al. (1998) or CIFAR-10 Krizhevsky (2009), the image datasets the three metrics were originally presented on, we hypothesize that it may not be effective with higher-dimensional inputs as observed in our study. Thus, it does not come as unexpected that methods like LRP are ranked high as they heavily filter the importance of pixels (in the case of LRP through relevance) compared to e.g. either VG, which

| Evaluation Criteria: | Faithfullness $(n_{(AM)} = 90)$ | | | Robustness $(n_{(AM)} = 63)$ | | | Complexity $(n_{(AM)} = 27)$ | | |
|---|---|---|---|---|---|---|---|---|---|
| Modality: | Image | Volume | Point Cloud | Image | Volume | Point Cloud | Image | Volume | Point Cloud |
| OC | 11.5 | 9 | 12 | 12.5 | 9.5 | 12 | 9 | 7.5 | 6.5 |
| LIME | 16.5 | 13 | 2 | 16 | 15 | 14 | 7 | 9.5 | 9 |
| KS | 16.5 | 12 | 3 | 17 | 17 | 12 | 12.5 | 12 | 10 |
| VG | 13 | 14 | 8.5 | 4 | 4 | 6.5 | 11 | 14 | 8 |
| IxG | 9.5 | 5 | 5 | 14.5 | 11.5 | 9.5 | 4 | 2 | 3 |
| GB | 7.5 | 10 | 1 | 6.5 | 8 | 12 | 5 | 6 | 5 |
| GC | 6 | 17 | - | 2.5 | 13 | - | 14 | 9.5 | - |
| SC | 4.5 | 11 | - | 10 | 15 | - | 10 | 16.5 | - |
| C+ | 11.5 | 15.5 | - | 5 | 11.5 | - | 15.5 | 16.5 | - |
| IG | 4.5 | 2 | 4 | 10 | 4 | 9.5 | 3 | 3.5 | 4 |
| EG | 1 | 3 | 7 | 2.5 | 1 | 6.5 | 17 | 7.5 | 11 |
| DL | 7.5 | 5 | 6 | 12.5 | 7 | 8 | 2 | 3.5 | 2 |
| DLS | 2 | 5 | 12 | 6.5 | 9.5 | 5 | 6 | 5 | 6.5 |
| LRP | 15 | 15.5 | 10 | 14.5 | 15 | 2 | 1 | 1 | 1 |
| RA | 14 | 8 | 8.5 | 1 | 6 | 3.5 | 8 | 11 | 12 |
| RoA | 9.5 | 7 | 12 | 8 | 2 | 3.5 | 15.5 | 13 | 14 |
| LA | 3 | 1 | 14 | 10 | 4 | 1 | 12.5 | 15 | 13 |

Per modality: Top 1 | Top 2-4 | Bottom 2-4 | Bottom 1

Table 2: Ranking of the average rank over model architectures, datasets, and all evaluation metrics of the respective criteria for each XAI method and modality (i.e. the rank of OC on an image is based on $3 * 3 * 10 = 90$ ranks). Coloring coincides with top and bottom positions as point cloud rankings are of length 14 and all others of length 17.

also acts on the pixel space but does not filter the gradients, or CAM methods, which attribute to local regions. If these XAI methods are less complex and more human understandable on computer vision modalities is debatable (see Appendix J for a more detailed discussion). Also, Bhatt et al. (2020) argues that the Complexity metric is better suited for tabular data. In contrast, attention methods are ranked commendable more robust, whereas KS and LIME are not. Faithfulness scores are still largely uniform distribution among the methods at this analysis level, indicating differences in faithfulness are caused by more distinguishing factors such as modalities.

## 3.2 Q2: Does the efficacy of XAI methods vary across different computer vision modalities?

While modalities in computer vision share properties such as locality, spatial structure, or associated feature descriptors, other properties such as dimensionality or representation (e.g. grid versus collection of points) differ. To quantify such an effect on XAI methods and identify clusters, we average ranks of XAI methods within modalities and evaluation criteria in Table 2, revealing some new distinctive patterns in the ranking of various methods. The extended table in Appendix H shows that ranking disparities between datasets within individual modalities are minimal, suggesting that no new insights could be gained by including more datasets. We observe from Table 2, that both LIME and KS consistently rank low across all three evaluation criteria except for point cloud faithfulness, where both methods achieve high rankings. Further, KS diverges considerably from the other two SHAP approximating methods, DLS and EG, which maintain consistently high rankings across the image and volume modalities, only declining on point clouds. These differences between SHAP approximating methods, but not in terms of XAI evaluation, are also demonstrated in recent research by Molnar et al. (2022). Regarding the attention methods, LA exhibits high faithfulness rankings specifically for the image and volume modalities, and all three attention methods are ranked high for robustness over all modalities. CAM methods, on the other hand, demonstrate average faithfulness and robustness within the image modality but rank considerably lower on the volume modality. Their complexity scores remain low across both modalities as indicated in subsection 3.1.

## 3.3 Q3: With the ascendency of Transformer architectures, is there a potential misalignment with established attribution-based XAI methods?

Specific architectural choices in Transformer architectures, such as negative activation outputs, matrix multiplication in attention layers, or skip connections can, in theory, affect attribution methods (see Chefer et al. (2021) for more details). To uncover such biases, we compare the difference in faithfulness rankings of attribution methods between CNN and Transformer architectures, as biased methods should be less faithful to the model (we refer to Appendix H for the full distinct versions

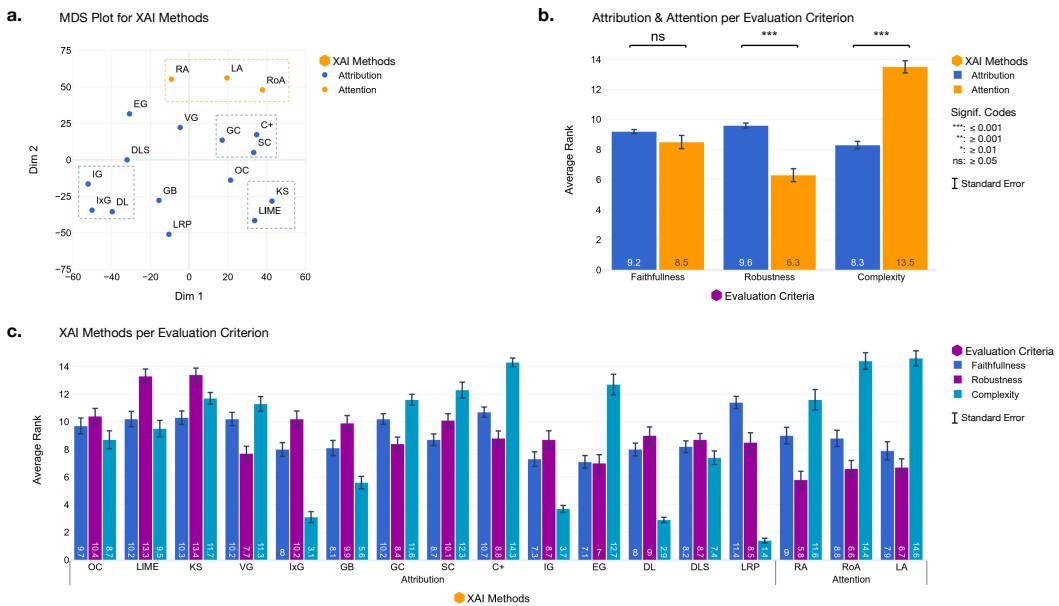

Figure 3: **a.** MDS plot of all rankings shows that similar operating methods are also clustered (thus ranked) similar, suggesting that the rankings are not random. **b.** Average rank per criteria shows significant (two-sample T-test) differences between attribution and attention methods for robustness and complexity but not faithfulness. **c.** Average rank per XAI method and evaluation criteria.

of Table 2 subdivided into CNN and Transformer architectures.). To this end, we compute the Kendals-$\tau$ rank correlation between each of the three architectures per dataset and compute their average correlation per modality (see Figure 4 (a.)). We observe a positive correlation between all rankings. For the point cloud modality, however, the correlation is significantly lower than for the other two modalities, indicating less similar rankings between model architectures. For volume and image modality, the similarity between CNN architectures is generally higher.

To examine the architecture's effect on image and volume data in more detail, we compute for each attribution method the rank difference between CNN and Transformer architectures (see Figure 4(b.)), as especially score backpropagation- and Deep Taylor Decomposition-based methods such as GB, DL, DLS, and LRP, but also CAM methods should be affected. Thus, if the method is, on average, ranked more faithful on CNN architectures, the difference in rank should be positive (the overall average is zero). The low ranking correlation across all architectures for point cloud data suggests that rank differences are primarily random, diminishing statistical power and indicating that patterns in point cloud rankings arise from factors other than model architectures. Indeed, if we compute the same figure for the point cloud architectures in Appendix I, we observe no significant difference from zero, except for OC, IxG, and LRP. In Figure 4(b.), we observe, as expected, worse faithfulness for GC, C+, DL, and DLS on Transformer architectures, while only DL and DLS are significantly different from zero and other hypothesized methods such as LRP or SC are not significantly different from zero. But other XAI methods, such as OC and IG, are significantly less faithful on Transformer architectures. Based on the presented results, we observe a trend that specific methods perform worse on Transformer architectures, but nothing that would significantly confirm the hypothesis.

## 3.4 MAIN INSIGHTS & TAKE-AWAYS

From the empirical analysis, we distill our findings into general and question-specific results. Practical recommendations from these results are outlined in the subsequent *(take-aways)*.

1. No XAI method ranks consistently high on all evaluation criteria. *(We recommend EG as the initial approach due to its reliable good performance in faithfulness and robustness, especially*

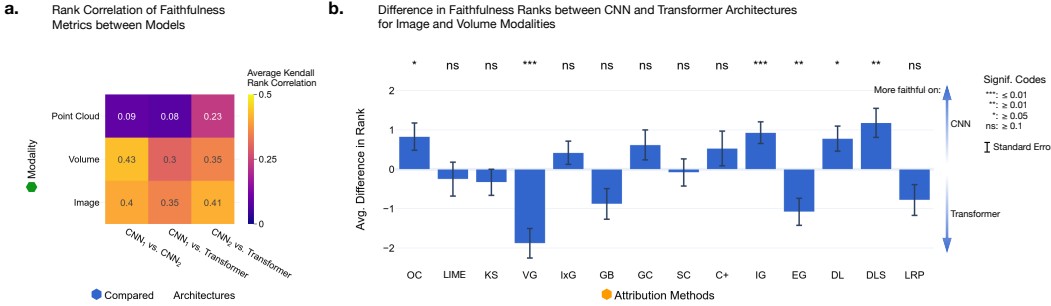

Figure 4: **a.** Kendall rank correlation between model architectures averaged over datasets and faithfulness criteria. **b.** Average difference in rank when subtracting the average rank of CNN architectures from the rank of Transformer architectures over faithfulness metrics. T-test to validate if the difference significantly differs from 0.

*for data with non-trivial to select baseline values.)*

2. Rankings of XAI methods generally generalize well over datasets given that observations have the same size. *(Focus on data properties such as dimensionality or modality rather than domains when selecting XAI methods for your problem.)*

3. Results in complexity seem counterintuitive, indicating that the evaluation objective of complexity metrics in computer vision does not always has to match the perception of low complexity. *(Given that CAM or attention methods, which appear less fine-grained, receive low complexity ranks, we recommend using these metrics cautiously, especially on high-dimensional data.)*

4. Answer to Q1: Attention methods are generally more robust and attribution methods less complex (see item 3), while differences in faithfulness depend on the specific method and modality. *(As relevance filtered attention (LA) consistently scores better than non-filtered raw attention, we would always prefer it.)*

5. Answer to Q2: We observe significant variations in evaluation performance manifest across modalities, especially for point cloud data. *(Given LIME and KS's poor performance on image and volume data, we advise against using them for high-dimensional and complex relationships.)*

6. Answer to Q3: Although we observe a trend that theoretically biased XAI methods are less faithful on Transformer architectures, the significance of these effects highly fluctuates between methods. *(As only DL and DLS stand out as performing significantly less faithfully on Transformer than on CNN architectures, we would not advise against any other attribution methods for Transformer architectures at this point.)*

## 4 COMPARISON WITH RELATED WORK

As we previously stated, the significance of current research is limited due to small and varying subsets in XAI methods and criteria used for evaluation, which consequently lead to contradicting outcomes. Due to the comprehensiveness of our study, we argue that the validity is significantly increased compared to previous work and that our findings offer a more definitive resolution to the inconsistencies observed in earlier, smaller-scale investigations. The difference in scale is particularly evident from Table 1, which presents a summary of 16 related relevant studies (all on image data), with an indication of what criteria were evaluated and how XAI methods performed relative to other methods or baselines analyzed in the study, based on our assessment. We present our results for the image modality indicated at the bottom, which do not necessarily transfer to other modalities as Table 2 shows. While some "evergreen" XAI methods i.e. VG, IxG, GB, and IG, stand out, Table 1 visualizes how sparse the field of XAI evaluation is, especially for attention methods.

In several cases, we observe similar results to other studies, e.g. Adebayo et al. (2018); Nie et al. (2018); Hooker et al. (2019); Yang & Kim (2019) critic on the unfaithfulness of GB and Bhatt et al. (2020); Yang & Kim (2019); Kakogeorgiou & Karantzalos (2021) evaluated IxG as unfaithful as

both methods perform partial input recovery. However, we observe that these results can differ between modalities, as GB, for example, is considerably faithful on point cloud data. Our faithfulness results especially contradict for GC with Arras et al. (2022); Chefer et al. (2021), LRP with Arras et al. (2022) and IG with Kakogeorgiou & Karantzalos (2021); Arun et al. (2021); Hooker et al. (2019). On the contrary, our observations are supported by Chefer et al. (2021) for LRP unfaithfulness, Chefer et al. (2021); Yang & Kim (2019); Arun et al. (2021); Kakogeorgiou & Karantzalos (2021) for GC faithfulness and Arras et al. (2022); Bhatt et al. (2020) for IG faithfulness, with Kindermans et al. (2019) arguing that the faithfulness of IG highly depends on the selected baseline. Noteworthy, our best-evaluated method in terms of faithfulness and robustness, EG, was never evaluated in any related study. Further, only Chefer et al. (2021) and Hesse et al. (2023) included attention methods, and both found LA to be the most faithful, a finding that our work supports.

The majority of quantitative studies evaluate faithfulness (10/15), compared to robustness (5/15) and complexity (2/15, Singh et al. (2021) evaluate qualitatively). Regarding robustness, Bhatt et al. (2020) and Yeh et al. (2019) rank KS as highly robust on lower dimensional image data. While we confirm the robustness of other SHAP methods, such as EG and DLS, KS is not robust in our image data evaluation. In terms of faithfulness, however, KS and LIME improve on the lower dimensional point cloud data, supporting the claim of Yeh et al. (2019) for KS and Zafar & Khan (2021) for LI, arguing that both methods have low faithfulness and robustness if the underlying model architecture or data is too complex to be modeled by a linear approximation on feature masks. The quantitative work in the field of complexity only includes papers that present a metric (Bhatt et al., 2020; Nguyen & Martínez, 2020; Kakogeorgiou & Karantzalos, 2021). We deemed the suitability of these metrics for computer vision modalities as debatable (see subsection 3.1). However, most work in complexity and human understandability is qualitative. As a notable example, Singh et al. (2021) showed that ophthalmologists and optometrists rate GB as a highly understandable method on the OCT dataset, claims that our quantitative work would support, in specific for the same OCT dataset, but e.g. Kakogeorgiou & Karantzalos (2021) heavily contradicts. We assess this high fluctuation between quantitative and especially qualitative complexity evaluation outcomes as further support for our hypothesis that there is a gap between the aim of the metrics and human conception of low complexity. In summary, our study aligns with several previous results, but especially faithful and robust evaluated methods such as EG and DLS are not present in previous work, and we observe discrepancies between modalities, as well as quantitative and qualitative evaluation of complexity.

## 5 CONCLUSION AND DISCUSSION

While we spent a significant amount of time adapting and parameterizing the evaluation metrics, we still encountered shortcomings during our experiments. Some metrics can be relatively sensitive to parameterization or perturbation baseline selections and depend on the properties of the dataset. We discuss these operational shortcomings in Appendix J. Even if a metric fails in specific situations, such as complexity metrics on higher-dimensional data, we still include them in the benchmark, as the determination of such failure cases is pivotal to future work in meta-evaluation.

Although our benchmark is one of the most comprehensive in the field, we restricted us to the modalities with the, in our opinion, most unique and not overlapping characteristics, ignoring e.g. video data, and did not include more unconventional post-hoc XAI methods such as symbolic representations or metamodels. Further, we did not include the evaluation criteria of localization and axiomatic properties as they either require ground-truth bounding boxes or can not be applied to all XAI methods. Finally, we want to comment that our benchmark focuses on the comparison between methods, not on the evaluation of whether a method is faithful or robust in general, thus ignoring e.g. synthetic baselines.

Our work was motivated by imminent challenges of the field, such as unreliable small-scale settings and the dismissing of key factors and new emerging methods. As a solution, we propose the LATEC benchmark, which not only serves as a more reliable platform for standardized large-scale evaluation but also allows to answer a multitude of questions pivotal to the field's advancement, from which we exemplary address three and distill the findings into main insights and take-aways. In the future, the benchmark can not only be used to evaluate new XAI methods but also leverage the vast amount of high-quality saliency maps to compare ranking behavior between evaluation metrics, providing a foundation for the emerging field of meta-evaluation.

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

APPENDIX

# A   LIST OF ABBREVIATIONS

## A.1   GENERAL

**XAI** . . . . . . . . . . . . . Explainable Artificial Intelligence
**LATEC** . . . . . . . . . Large-scale Attribution & Attention Evaluation in Computer Vision
**DL** . . . . . . . . . . . . . . Deep Learning
**CV** . . . . . . . . . . . . . . Computer Vision
**CNN** . . . . . . . . . . . . Convolutional Neural Network
**KNN** . . . . . . . . . . . . k-Nearest Neighbor
**MDS** . . . . . . . . . . . . Multidimensional Scaling

## A.2   DATASETS

**IMN** . . . . . . . . . . . . ImageNet
**OCT** . . . . . . . . . . . . UCSD OCT Retina
**R45** . . . . . . . . . . . . . RESISC45 - Remote Sensing Image Scene Classification
**AMN** . . . . . . . . . . . . Adrenal-MedMNIST3D
**OMN** . . . . . . . . . . . Organ-MedMNIST3D
**VMN** . . . . . . . . . . . . Vessel-MedMNIST3D
**CMA** . . . . . . . . . . . . CoMA - Generating 3D faces using Convolutional Mesh Autoencoders
**M40** . . . . . . . . . . . . . ModelNet40
**SHN** . . . . . . . . . . . . ShapeNet

## A.3   XAI METHODS

**OC** . . . . . . . . . . . . . . Occlusion
**LIME** . . . . . . . . . . . Local Interpretable Model-Agnostic Explanations
**KS** . . . . . . . . . . . . . . Kernel SHAP
**VG** . . . . . . . . . . . . . . Vanilla Gradient
**IxG** . . . . . . . . . . . . . Input x Gradient
**GB** . . . . . . . . . . . . . . Guided Backprob
**GC** . . . . . . . . . . . . . . GradCAM
**SC** . . . . . . . . . . . . . . ScoreCAM
**C+** . . . . . . . . . . . . . . GradCAM++
**IG** . . . . . . . . . . . . . . Integrated Gradients
**EG** . . . . . . . . . . . . . Expected Gradients
**DL** . . . . . . . . . . . . . . DeepLIFT
**DLS** . . . . . . . . . . . . DeepLIFT SHAP
**LRP** . . . . . . . . . . . . Layer-Wise Relevance Propagation
**RA** . . . . . . . . . . . . . . Raw Attention
**RoA** . . . . . . . . . . . . Rollout Attention
**LA** . . . . . . . . . . . . . . LRP Attention

# B  DEEP LEARNING MODEL PERFORMANCE AND HYPERPARAMETER

## B.1  TEST SET PERFORMANCE

**a.**  Testset Performance on Image Modality

| Dataset: | Model Architecture: | Model Performance Metric | | | | |
|---|---|---|---|---|---|---|
| | | Accuracy | Precision | Recall | F1 | AUROC |
| OCT (MC: 4) | ResNet 50 | 0.999 | 0.999 | 0.999 | 0.999 | 1.0 |
| | EfficientNet b0 | 0.9969 | 0.9969 | 0.9969 | 0.9969 | 1.0 |
| | DeiT ViT | 0.999 | 0.999 | 0.999 | 0.999 | 1.0 |
| R45 (MC: 45) | ResNet 50 | 0.9535 | 0.9536 | 0.9538 | 0.9535 | 0.9995 |
| | EfficientNet b0 | 0.9554 | 0.9554 | 0.9549 | 0.9549 | 0.9995 |
| | DeiT ViT | 0.9568 | 0.957 | 0.9568 | 0.9567 | 0.9995 |

**b.**  Testset Performance on Volume Modality

| Dataset: | Model Architecture: | Model Performance Metric | | | | |
|---|---|---|---|---|---|---|
| | | Accuracy | Precision | Recall | F1 | AUROC |
| AMN (BC) | 3D ResNet 18 | 0.8003 | 0.8013 | 0.7987 | 0.8 | 0.8699 |
| | EfficientNet3D b0 | 0.8003 | 0.7954 | 0.8087 | 0.802 | 0.8647 |
| | Simple3DFormer | 0.7936 | 0.7907 | 0.7907 | 0.7907 | 0.8728 |
| OMN (MC: 11) | 3D ResNet 18 | 0.9115 | 0.9248 | 0.9248 | 0.9226 | 0.9953 |
| | EfficientNet3D b0 | 0.8754 | 0.8924 | 0.8936 | 0.8914 | 0.9893 |
| | Simple3DFormer | 0.8131 | 0.8463 | 0.8381 | 0.84 | 0.9815 |
| VMN (BC) | 3D ResNet 18 | 0.9359 | 0.937 | 0.9346 | 0.9358 | 0.98 |
| | EfficientNet3D b0 | 0.9162 | 0.9162 | 0.9162 | 0.9162 | 0.9229 |
| | Simple3DFormer | 0.8861 | 0.8871 | 0.8848 | 0.886 | 0.9394 |

**c.**  Testset Performance on Point Cloud Modality

| Dataset: | Model Architecture: | Model Performance Metric | | | | |
|---|---|---|---|---|---|---|
| | | Accuracy | Precision | Recall | F1 | AUROC |
| CMA (MC: 12) | PointNet | 0.9852 | 0.9743 | 0.9876 | 0.98 | 0.998 |
| | DGCNN | 0.9535 | 0.9373 | 0.9498 | 0.9423 | 0.9989 |
| | PC Transformer | 0.9751 | 0.9645 | 0.9688 | 0.9662 | 0.9996 |
| M40 (MC: 40) | PointNet | 0.8914 | 0.8374 | 0.8564 | 0.8438 | 0.9958 |
| | DGCNN | 0.9177 | 0.8844 | 0.891 | 0.8864 | 0.9973 |
| | PC Transformer | 0.9149 | 0.8779 | 0.8842 | 0.8796 | 0.9969 |
| SHN (MC: 16) | PointNet | 0.9878 | 0.9673 | 0.9689 | 0.9668 | 0.9991 |
| | DGCNN | 0.9903 | 0.966 | 0.9847 | 0.9745 | 0.9995 |
| | PC Transformer | 0.9896 | 0.9642 | 0.9819 | 0.9716 | 0.9997 |

**MC #**: Multi-Class (# Classes), **BC**: Binary-Class

Table 3: **a., b.  & c.** Test set performance measured with the metrics: accuracy, precision, recall, F1, and area under the receiver operating characteristic (AUROC) curve, for each modality. In the case of IMN we use pretrained weights for the Transformer architecture from Huggingface[1] and the CNN architectures from TorchHub[2,3].

[1] https://huggingface.co/facebook/deit-small-patch16-224
[2] https://pytorch.org/vision/stable/models/generated/torchvision.models.resnet50.html
[3] https://pytorch.org/vision/stable/models/generated/torchvision.models.efficientnet_b0.html

Architectures were chosen based on their popularity and, to a limited extent, comparability between modalities, e.g. ResNet-50 and 3D ResNet-18 which both emerge from the same family of ResNet architectures. While 3D volume architectures could also be applied to point cloud data, we choose point cloud specific architectures for the modality.

## B.2 HYPERPARAMETER

We tuned all hyperparameters on either the declared validation set or sampled a validation set based on 20% of the train set. The tuning was performed via grid search for each model. The primary metric for hyperparameter tuning was the F1 score.

**a.**   Hyperparameter for Image Modality

| | | Utilized Computer Vision Datasets | |
|---|---|---|---|
| Model Architecture: | Hyperparameter: | OCT | R45 |
| ResNet 50 | Batch size | 128 | 128 |
| | Max Epochs | 8 | 60 |
| | Learning rate (LR) | 0.0001 | 0.0001 |
| | Optimizer | Madgrad | Madgrad |
| | LR Scheduler | Cosine Annealing | Cosine Annealing |
| | Weight Decay | 0 | 0 |
| | Momentum | 0.9 | 0.9 |
| | Augmentations | Train:
Resize (256,256)
RandomCrop (224,224)
RandomAffine (shear=0.2, degrees=5)
RandomHorizontalFlip
Grayscale (channels=3)

Test:
Resize (256,256)
CenterCrop (224,224)
Grayscale (channels=3) | Train:
Resize (256,256)
RandomCrop (224,224)
RandomHorizontalFlip
RandAugment
Normalize (mean=(0.485, 0.456, 0. 406), std=(0.229, 0.224, 0.225))

Test:
Resize (256,256)
Normalize (mean=(0.485, 0.456, 0. 406), std=(0.229, 0.224, 0.225)) |
| | Sampling | Weighted Random Sampling | None |
| EfficientNet b0 | Batch size | 128 | 128 |
| | Max Epochs | 5 | 15 |
| | Learning rate (LR) | 0.0001 | 0.001 |
| | Optimizer | Madgrad | Madgrad |
| | LR Scheduler | Cosine Annealing | Cosine Annealing |
| | Weight Decay | 0 | 0 |
| | Momentum | 0.9 | 0.9 |
| | Augmentations | Train:
Resize (256,256)
RandomCrop (224,224)
RandomAffine (shear=0.2, degrees=5)
RandomHorizontalFlip
Grayscale (channels=3)

Test:
Resize (256,256)
CenterCrop (224,224)
Grayscale (channels=3) | Train:
Resize (256,256)
RandomCrop (224,224)
RandomHorizontalFlip
RandAugment
Normalize (mean=(0.485, 0.456, 0. 406), std=(0.229, 0.224, 0.225))

Test:
Resize (256,256)
Normalize (mean=(0.485, 0.456, 0. 406), std=(0.229, 0.224, 0.225)) |
| | Sampling | Weighted Random Sampling | None |
| DeIT ViT | Batch size | 128 | 128 |
| | Max Epochs | 6 | 60 |
| | Learning rate (LR) | 0.0001 | 0.0001 |
| | Optimizer | Madgrad | Madgrad |
| | LR Scheduler | Cosine Annealing | Cosine Annealing |
| | Weight Decay | 0 | 0 |
| | Momentum | 0.9 | 0.9 |
| | Augmentations | Train:
Resize (256,256)
RandomCrop (224,224)
RandomAffine (shear=0.2, degrees=5)
RandomHorizontalFlip
Grayscale (channels=3)

Test:
Resize (256,256)
CenterCrop (224,224)
Grayscale (channels=3) | Train:
Resize (256,256)
RandomCrop (224,224)
RandomHorizontalFlip
RandAugment
Normalize (mean=(0.485, 0.456, 0. 406), std=(0.229, 0.224, 0.225))

Test:
Resize (256,256)
Normalize (mean=(0.485, 0.456, 0. 406), std=(0.229, 0.224, 0.225)) |
| | Sampling | Weighted Random Sampling | None |

Table 4: Hyperparameter for all three architectures and CV datasets, excluding IMN as we load pretrained weights.

**a.**    Hyperparameter for Volume Modality

| Model Architecture: | Hyperparameter: | Utilized Computer Vision Datasets | | |
|---|---|---|---|---|
| | | AMN | OMN | VMN |
| 3D ResNet18 | Batch size | 32 | 32 | 32 |
| | Max Epochs | 100 | 100 | 100 |
| | Learning rate (LR) | 0.001 | 0.001 | 0.001 |
| | Optimizer | SGD | Adam | Adam |
| | LR Scheduler | Cosine Annealing | Cosine Annealing | Cosine Annealing |
| | Weight Decay | 0 | 0 | 0 |
| | Momentum | 0.9 | 0 | 0 |
| | Augmentations | Train: RandomBrightness($U(0,1)$)  Test: FixedBrightness(0.5) | None | Train: RandomBrightness($U(0,1)$)  Test: FixedBrightness(0.5) |
| | Sampling | Weighted Random Sampling | None | Weighted Random Sampling |
| 3D EfficientNet b0 | Batch size | 32 | 32 | 64 |
| | Max Epochs | 100 | 100 | 100 |
| | Learning rate (LR) | 0.001 | 0.001 | 0.001 |
| | Optimizer | SGD | AdamW | Adam |
| | LR Scheduler | Cosine Annealing | Cosine Annealing | Cosine Annealing |
| | Weight Decay | 0.0005 | 0.0005 | 0 |
| | Momentum | 0.9 | 0 | 0 |
| | Augmentations | Train: RandomBrightness($U(0,1)$)  Test: FixedBrightness(0.5) | None | Train: RandomBrightness($U(0,1)$)  Test: FixedBrightness(0.5) |
| | Sampling | Weighted Random Sampling | None | Weighted Random Sampling |
| Simple3DFormer | Batch size | 32 | 32 | 64 |
| | Max Epochs | 150 | 100 | 100 |
| | Learning rate (LR) | 0.001 | 0.000001 | 0.001 |
| | Optimizer | SGD | Madgrad | Adam |
| | LR Scheduler | Cosine Annealing | Cosine Annealing | Cosine Annealing |
| | Weight Decay | 0.0005 | 0 | 0 |
| | Momentum | 0.9 | 0.9 | 0 |
| | Augmentations | Train: RandomBrightness($U(0,1)$)  Test: FixedBrightness(0.5) | None | Train: RandomBrightness($U(0,1)$)  Test: FixedBrightness(0.5) |
| | Sampling | Weighted Random Sampling | None | Weighted Random Sampling |

Table 5: Hyperparameters for all three architectures and CV datasets.

**a.** Hyperparameter for Point Cloud Modality

| Model Architecture: | Hyperparameter: | Utilized Computer Vision Datasets | | |
|---|---|---|---|---|
| | | CMA | M40 | SHN |
| PointNet | Batch size | 32 | 24 | 32 |
| | Max Epochs | 100 | 200 | 200 |
| | Learning rate (LR) | 0.001 | 0.001 | 0.001 |
| | Optimizer | AdamW | AdamW | AdamW |
| | LR Scheduler | Cosine Annealing | Cosine Annealing | Cosine Annealing |
| | Weight Decay | 0.0001 | 0.0001 | 0.0001 |
| | Momentum | 0 | 0 | 0 |
| | Augmentations | Pretransforms:
NormalizeScale

Train:
SamplePoints (1024)
RandomScale (0.67,1.5)
RandomRotate (degrees=15)
RandomJitter (0.02)

Test:
SamplePoints (1024) | Pretransforms:
NormalizeScale

Train:
SamplePoints (1024)
RandomScale (0.67,1.5)
RandomJitter (0.02)

Test:
SamplePoints (1024) | Pretransforms:
NormalizeScale

Train:
RandomScale (0.67,1.5)
RandomJitter (0.01)
RandomRotate(degress=15,
axis = (0,1,2))

Test:
None |
| | Sampling | None | None | None |
| DGCNN | Batch size | 32 | 32 | 32 |
| | Max Epochs | 100 | 250 | 200 |
| | Learning rate (LR) | 0.001 | 0.001 | 0.001 |
| | Optimizer | AdamW | AdamW | AdamW |
| | LR Scheduler | Cosine Annealing | Cosine Annealing | Cosine Annealing |
| | Weight Decay | 0.0001 | 0.0001 | 0.0001 |
| | Momentum | 0 | 0 | 0 |
| | Augmentations | Pretransforms:
NormalizeScale

Train:
SamplePoints (1024)
RandomScale (0.67,1.5)
RandomRotate (degrees=15)
RandomJitter (0.02)

Test:
SamplePoints (1024) | Pretransforms:
NormalizeScale

Train:
SamplePoints (1024)
RandomScale (0.67,1.5)
RandomJitter (0.02)

Test:
SamplePoints (1024) | Pretransforms:
NormalizeScale

Train:
RandomScale (0.67,1.5)
RandomJitter (0.01)
RandomRotate(degress=15,
axis = (0,1,2))

Test:
None |
| | Sampling | None | None | None |
| PC Transformer | Batch size | 32 | 32 | 32 |
| | Max Epochs | 150 | 250 | 200 |
| | Learning rate (LR) | 0.01 | 0.01 | 0.01 |
| | Optimizer | SGD | SGD | SGD |
| | LR Scheduler | Cosine Annealing | Cosine Annealing | Cosine Annealing |
| | Weight Decay | 0.0005 | 0.0005 | 0.0005 |
| | Momentum | 0.9 | 0.9 | 0.9 |
| | Augmentations | Pretransforms:
NormalizeScale

Train:
SamplePoints (1024)
RandomScale (0.67,1.5)
RandomRotate (degrees=15)
RandomJitter (0.02)

Test:
SamplePoints (1024) | Pretransforms:
NormalizeScale

Train:
SamplePoints (1024)
RandomScale (0.67,1.5)
RandomJitter (0.02)

Test:
SamplePoints (1024) | Pretransforms:
NormalizeScale

Train:
RandomScale (0.67,1.5)
RandomJitter (0.01)
RandomRotate(degress=15,
axis = (0,1,2))

Test:
None |
| | Sampling | None | None | None |

Table 6: Hyperparameters for all three architectures and CV datasets.

## C  XAI METHODS OVERVIEW, PARAMETERS, AND ADAPTION

### C.1  OVERVIEW

#### C.1.1  ATTRIBUTION METHODS

**Occlusion [OC]** (Zeiler & Fergus, 2013)    Systematically obscures different parts of the input data and observes the resulting impact on the output, to determine which parts of the data are most important for the model's predictions.

**LIME [LIME]** (Ribeiro et al., 2016)    Creates an interpretable model around the prediction of a complex model to explain individual predictions locally (patch-based in our case), using perturbations of the input data and observing the corresponding changes in the output.

**Kernel SHAP [KS]** (Lundberg & Lee, 2017)    Using a weighted linear regression model as the local surrogate and selecting a suitable weighting kernel, the regression coefficients from the LIME surrogate can estimate the SHAP values.

**Vanilla Gradient [VG]** (Simonyan et al., 2013)    The raw input gradients of the model.

**Input x Gradient [IxG]** (Shrikumar et al., 2017)    Multiples the input features by their corresponding gradients with respect to the model's output.

**Guided Backprob [GB]** (Springenberg et al., 2015)    Modifies the standard backpropagation process to only propagate positive gradients for positive inputs through the network, thereby creating visualizations that highlight the features that strongly activate certain neurons in relation to the target output.

**GradCAM [GC]** (Selvaraju et al., 2017)    Uses the gradients of the target class flowing into the final convolutional layer to produce a coarse localization map by, highlighting the important regions in the image by up-scaling the map.

**ScoreCAM [SC]** (Wang et al., 2020)    Eliminates the need for gradient information by determining the importance of each activation map based on its forward pass score for the target class, producing the final output through a weighted sum of these activation maps.

**GradCAM++ [C+]** (Chattopadhay et al., 2018)    Generates a visual explanation for a given class label by employing a weighted sum of the positive partial derivatives from the final convolutional layer's feature maps, using them as weights with respect to the class score.

**Integrated Gradients [IG]** (Sundararajan et al., 2017)    Explains model predictions by attributing the prediction to the input features, calculating the path integral of the gradients along the straight-line path from a baseline input to the actual input.

**Expected Gradients [EG]** (Erion et al., 2020)    Also called Gradient SHAP. Avoids the selection of a baseline value compared to IG, by leveraging a probabilistic baseline computed over a sample of observations.

**DeepLIFT [DL]** (Shrikumar et al., 2017)    Assigns contribution scores to each input feature based on the difference between the feature's activation and a reference activation, effectively measuring the feature's impact on the output compared to a baseline.

**DeepLIFT SHAP [DLS]** (Lundberg & Lee, 2017)    Cmbines the DeepLIFT method with Shapley values to assign importance scores to input features by computing their contributions to the output relative to a reference input, while ensuring consistency with Shapley values.

**Layer-Wise Relevance Propagation [LRP]** (Binder et al., 2016)    Explains neural network decisions by backpropagating the output prediction through the layers, redistributing relevance scores to the input features to visualize their contribution to the final decision. We use the $\epsilon$-,$\gamma$- and $0^+$-rules depending on the model architecture for relevance backpropagation.

### C.1.2   ATTENTION METHODS

**Raw Attention [RA]** (Dosovitskiy et al., 2021)    Rearranged and up-scaled attention values of the last attention head.

**Rollout Attention [RoA]** (Abnar & Zuidema, 2020)    Averages attention weights of multiple heads to trace the contribution of each part of the input data through the network.

**LRP Attention [LA]** (Chefer et al., 2021)    Assigns local relevance scores to attention weights based on the Deep Taylor Decomposition principle and propagates these relevancy scores through the model.

## C.2 PARAMETERS

| XAI Method: | OC | | | | LIME | | | KS | | | CAM (all) | SC | IG | | EG | | DL | | LRP | | | RA |
|---|---|---|---|---|---|---|---|---|---|---|---|---|---|---|---|---|---|---|---|---|---|---|
| Parameter: | strides | sliding_window_shapes | baseline | perturbations_per_eval | alpha | n_samples | perturbations_per_eval | baseline | n_samples | perturbations_per_eval | layer | batch_size | baseline | n_steps | n_samples | std | eps | baseline | rule | eps | gamma | layer |
| Image | 25 | (50, 50) | 0 | 1 | 1,0 | 10 | 5 | 0 | 10 | 5 | ResNet50,layer4[-1] EfficientNetbo,features[-1] ViT,blocks[-1],norm1 | 32 | 0 | 30 | 40 | 0,001 | 1e-9 | 0 | ε & γ-rule / 0+-rule | 0,0001 | 0,25 | ViT,blocks[-1],attn |
| Volume | 4 | (7, 7, 7) | 0 | 1 | 1,0 | 10 | 5 | 0 | 10 | 5 | 3DEfficientNetbo,blocks[-13] 3DResNet18,layer3 S3DF,blocks[-1],norm1 | 64 | 0 | 30 | 40 | 0,001 | 1e-9 | 0 | ε & γ-rule / 0+-rule | 0,0001 | 0,25 | S3DF[-1],attn |
| Point Cloud | 1 | (3,1) | 0 | 5 | 4,0 | 10 | 5 | 0 | 10 | 5 | PointNet,transform,bn1 DGCNN,conv5 PCT,sa4,after_norm | 16 | 0 | 30 | 16 | 0,001 | 1e-9 | 0 | ε & γ-rule / 0+-rule | 0,00001 | 0,25 | PCT,sa4,attn |

Table 7: Parameters for each XAI method and modality.

The parameters for each XAI method are derived for each modality via qualitative evaluation which we deem the most realistic scenario. We tuned the XAI methods on five observations per dataset and modality, which we argue is a fair trade-off between fitting the methods to the dataset but not overfitting them to bias the evaluation. We did not tune the parameters per dataset, as the parameters transfer very well between datasets and only needed minimal adjustments.

## C.3 ADAPTION

In this section, we explain how we adapted XAI methods in our framework to seamlessly work with 3D modalities. We neglect the methods that did not need any adaption (besides e.g. unit tests etc.) as they work independently of the input dimensions. All XAI methods are adapted, such that they only return positive attribution.

**Occlusion**     For the 3D modalities we implemented a 3D kernel as the perturbation baseline for volumes and a 1x3 mask (one point) for the point clouds. The image and volume mask transverse with overlap and the point cloud mask without overlap over all dimensions of the input object.

**LIME & Kernel SHAP**     For both methods, we implemented feature masks for each modality, as training the linear surrogate models on the original input features is not informative and computationally very expensive. Each mask groups the input features to the same interpretable feature. We use predefined grids as feature masks, as superpixel computing algorithms are too computational and time-expensive, especially for 3D modalities and evaluation metrics that perturb the input space or refit the XAI method multiple times. For the image modality, we use a 16x16x3, for volume 7x7x7, and for point cloud 1x3 (one point) mask, which is distributed as a non-overlapping grid in all dimensions over the whole object. For point clouds we use ridge regression and for the other modalities lasso regression.

**GradCAM, ScoreCAM & GradCAM++**     For all CAM methods on volume data we adapted the gradient averaging and the subsequent weighting of the activations and used nearest-neighbor interpolation to upscale the weighted activations to 3D volumes. In the case of ScoreCAM we also use nearest neighbor up-sampling instead of bilinear up-sampling, to upscale the activations for weighting the output of the previous layer. To correctly reshape the upscaled images and volumes in the case of the Transformer architectures (taking the channels to the first dimension as for CNNs), we use two different reshape functions for images and volumes when the CAM methods are applied to Transformer architectures. Further, we use the absolute activation output, not the non-negative for Transformer architectures, as the leaky-ReLU/GeLU function output otherwise would sometimes be zero.

**LRP**     For CNNs, we assigned the $\epsilon$-rule to the linear or identity layers, the identity rule to all non-linear layers, and to all other layers (convolutions, pooling, batch normalization, etc.) the $\gamma$-rule. For Transformer architectures we implemented the $0^{+}$-rule for all layers. However, for the Simple3DFormer and the PC Transformer, we had to add custom relevance propagation through the whole model, as the architectures come with several sub-modules such as "local gathering" for the PC Transformer, which are non-trivial to backpropagate through.

**Raw Attention**     We always use the raw attention of the last Transformer block and use bilinear or trilinear interpolation to rescale the attention for image and volume data. For point cloud data, this procedure is more complicated as the PC Transformer projects the embeddings on which the

Transformer acts via farthest point sampling and k-nearest neighbor grouping. Thus in each down-sampling step, we save which k points are sampled to then use k-nearest neighbor interpolation to cast the attention values for these remaining points back into the input space onto all 1024 original points.

**Rollout Attention**     Same procedure as for Raw Attention but before we interpolate back into the original input space, we use the rollout attention aggregation algorithm over all Transformer modules in the architecture.

**LRP Attention**     As for LRP we use custom relevance backpropagation for the Simple3DFormer and PC Transformer architectures. Based on the relevance scores, we filter the attention of each Transformer module, aggregate the filtered attention with the rollout algorithm, and interpolate the resulting attention back into the input as described for Raw Attention.

# D    EVALUATION METRICS OVERVIEW, PARAMETERS, AND ADAPTION

## D.1    OVERVIEW

### D.1.1    FAITHFULNESS

**Faithfulness Correlation** (Bhatt et al., 2020)     Gauges an explanation's fidelity to model behavior. It measures the linear correlation between predicted logits of modified test points and the average explanation for selected features, returning a score between -1 and 1. For each test, selected features are replaced with baseline values, and Pearson's correlation coefficient is determined, averaging results over multiple tests.

**Faithfulness Estimate** (Alvarez Melis & Jaakkola, 2018)     Evaluates the accuracy of estimated feature relevances by using a proxy for the "true" influence of features, as the actual influence is often unavailable. This is done by observing how the model's prediction changes when certain features are removed or obscured. Specifically, for probabilistic classification models, the metric looks at how the probability of the predicted class drops when features are removed. This drop is then compared to the interpreter's prediction of that feature's relevance. The metric also computes correlations between these probability drops and relevance scores across various data points.

**Monotonicity Correlation** (Nguyen & Martínez, 2020)     Evaluates the correlation between the absolute values of attributions and the uncertainty in probability estimation using Spearman's coefficient. If attributions are not monotonic the authors argue that they are not providing the correct importance of the features.

**Pixel Flipping** (Bach et al., 2015)     The core concept involves flipping pixels with very high, very low, or near-zero attribution scores. The effect of these changes is then assessed on the prediction scores, with the average prediction being determined.

**Region Perturbation** (Samek et al., 2017)     A step-by-step method where the class representation in the image, as determined by a function, diminishes as we gradually eliminate details from an image. This process, known as region perturbation, occurs at designated locations. Finally, the effect on the average prediction is calculated.

**Insertion** (Petsiuk et al., 2018)     Gradually inserts features into a baseline input, which is a strongly blurred version of the image, to not create OOD examples. During this process, the change in prediction is measured and the correlation with the respective attribution value is calculated.

**Deletion** (Petsiuk et al., 2018)     Deletes input features one at a time by replacing them with a baseline value based on their attribution score. During this process, the change in prediction is measured and the correlation with the respective attribution value is calculated.

**Iterative Removal of Features (IROF)** (Rieger & Hansen, 2020)     The metric calculates the area under the curve for each class based on the sorted average importance of feature segments (superpixels). As these segments are progressively removed and prediction scores gathered, the results are averaged across multiple samples.

**Remove and Debias (ROAD)** (Rong et al., 2022)     Evaluates the model's accuracy on a sample set during each phase of an iterative process where the k most attributed features are removed.

To eliminate bias, in every step, the k most significant pixels, by the most relevant first order, are substituted with noise-infused linear imputations.

**Sufficiency** (Dasgupta et al., 2022)    Assesses the likelihood that the prediction label for a specific observation matches the prediction labels of other observations which have similar saliency maps.

### D.1.2    ROBUSTNESS

**Local Lipschitz Estimate** (Alvarez Melis & Jaakkola, 2018)    Lipschitz continuity in calculus is a concept that measures the relative changes in a function's output concerning its input. While the traditional definition of Lipschitz continuity is global, focusing on the largest relative deviations across the entire input space, this global perspective isn't always meaningful in XAI. This is because expecting consistent explanations for vastly different inputs isn't realistic. Instead, a more localized approach, focusing on stability for neighboring inputs, is preferred, resulting in a pointwise, neighborhood-based local Lipschitz continuity metric.

**Max Sensitivity** (Yeh et al., 2019)    Measures the largest shift in the explanation when the input is slightly altered. It specifically evaluates the utmost sensitivity of a saliency map by taking multiple samples from a defined L-infinity ball subspace with a set input neighborhood radius, using Monte Carlo sampling for approximation.

**Continuity** (Montavon et al., 2018)    Evaluates, that if two observations are nearly equivalent, then the explanations of their predictions should also be nearly equivalent. It then measures the strongest variation of the explanation in the input domain.

**Relative Input/Output/Representation Stability** (Agarwal et al., 2022)    All metrics leverage model information to evaluate the stability of a saliency map with respect to the change in the either, input data, intermediate representations, and output logits of the underlying prediction model.

**Infidelity** (Yeh et al., 2019)    Calculates the expected mean-squared error (MSE) between the saliency map multiplied by a random variable input perturbation and the differences between the model at its input and perturbed input.

### D.1.3    COMPLEXITY

**Sparseness** (Chalasani et al., 2020)    Measures the Gini Index on the vector of absolute saliency map values. The assessment ensures that features genuinely influencing the output have substantial contributions, while insignificant or only slightly relevant features should have minimal contributions.

**Complexity** (Bhatt et al., 2020)    Determines the entropy of the normalized saliency map.

**Effective Complexity** (Nguyen & Martínez, 2020)    Evaluates the number of absolute saliency map values that surpass a threshold. Values above this threshold suggest the features are significant, while those below indicate they are not.

### D.2    PARAMETERS

We tuned the parameters of the evaluation metrics per dataset based on the distribution of their scores from Appendix K. We applied the suggested parameters from Hedström et al. (2022) or the respective papers. If the resulting score distributions were collapsed, almost uniform, or too indistinguishable between the XAI methods, we tuned the respective parameters. This step was completed prior to the ranking analysis, and no adjustments were made to the metrics once the ranking phase commenced.

| Evaluation Metric: | Parameter: | Image | | | Voxel | | | Point Cloud | | |
|---|---|---|---|---|---|---|---|---|---|---|
| | | IMN | OCT | R45 | AMN | OMN | VMN | CMA | M40 | SHN |
| Faithfulness Correlation | nr_runs | 100 | 100 | 100 | 100 | 100 | 100 | 100 | 100 | 100 |
| | subset_size | 224 | 224 | 224 | 56 | 56 | 56 | 32 | 32 | 32 |
| | perturb_baseline | black | black | black | black | black | black | center | center | center |
| Faithfulness Estimate | features_in_step | 224 | 224 | 224 | 56 | 56 | 56 | 32 | 32 | 32 |
| | perturb_baseline | black | black | black | black | black | black | center | center | center |
| Monotonicity Correlation | nr_samples | 10 | 10 | 10 | 10 | 10 | 10 | 10 | 10 | 10 |
| | features_in_step | 3136 | 3136 | 3136 | 392 | 392 | 392 | 256 | 256 | 256 |
| | perturb_baseline | uniform | uniform | uniform | uniform | uniform | uniform | uniform | uniform | uniform |
| Pixel Flipping | features_in_step | 224 | 224 | 224 | 56 | 56 | 56 | 32 | 32 | 32 |
| | perturb_baseline | black | black | black | black | black | black | center | center | center |
| Region Perturbation | patch_size | 14 | 14 | 18 | 4 | 4 | 4 | 3 | 3 | 3 |
| | regions_evaluation | 10 | 10 | 20 | 20 | 20 | 20 | 32 | 32 | 32 |
| | perturb_baseline | uniform | uniform | uniform | uniform | uniform | uniform | uniform | uniform | uniform |
| Insertion | pixel_batch_size | 50 | 50 | 50 | 50 | 50 | 50 | 50 | 50 | 50 |
| | sigma | 5.0 | 120.0 | 40.0 | 2.5 | 2.5 | 2.5 | 0.05 | 0.1 | 0.05 |
| | kernel_size | 15 | 39 | 19 | 1 | 1 | 1 | 1 | 1 | 1 |
| Deletion | pixel_batch_size | 50 | 50 | 50 | 50 | 50 | 50 | 50 | 50 | 50 |
| IROF | segmentation | Slic | Slic | Slic | 3D Slic | 3D Slic | 3D Slic | KMeans | KMeans | KMeans |
| | perturb_baseline | mean | mean | mean | black | black | black | center | center | center |
| ROAD | noise | 0.1 | 0.1 | 0.1 | 4.0 | 2.5 | 50.0 | 0.02 | 0.15 | 0.3 |
| | percentages_max | 100 | 100 | 100 | 100 | 100 | 100 | 100 | 100 | 100 |
| Sufficiency | threshold | 0.9 | 0.6 | 0.6 | 0.02 | 0.75 | 0.0002 | 0.75 | 0.75 | 0.6 |
| Local Lipschitz Estimate | nr_samples | 5 | 5 | 5 | 10 | 10 | 10 | 5 | 5 | 5 |
| | perturb_std | 0.1 | 0.0002 | 0.1 | 0.2 | 0.2 | 0.2 | 0.1 | 0.1 | 0.1 |
| | perturb_mean | 0.0 | 0.0 | 0.0 | 0.0 | 0.0 | 0.0 | 0.0 | 0.0 | 0.0 |
| MaxSensitivity | nr_samples | 10 | 10 | 10 | 10 | 10 | 10 | 10 | 10 | 10 |
| | lower_bound | 0.2 | 0.2 | 0.2 | 0.2 | 0.2 | 0.2 | 0.2 | 0.2 | 0.2 |
| Continuity | patch_size | 56 | 56 | 56 | 7 | 7 | 7 | 3 | 3 | 3 |
| | nr_steps | 20 | 20 | 20 | 20 | 20 | 20 | 20 | 20 | 20 |
| | perturb_baseline | uniform | uniform | uniform | uniform | uniform | uniform | uniform | uniform | uniform |
| RIS | nr_samples | 10 | 10 | 10 | 10 | 10 | 10 | 10 | 10 | 10 |
| ROS | nr_samples | 10 | 10 | 10 | 10 | 10 | 10 | 10 | 10 | 10 |
| RRS | nr_samples | 10 | 10 | 10 | 10 | 10 | 10 | 10 | 10 | 10 |
| Infidelity | n_perturb_samples | 50 | 50 | 50 | 50 | 50 | 50 | 50 | 50 | 50 |
| Effective Complexity | eps | 0.01 | 0.01 | 0.01 | 0.001 | 0.001 | 0.001 | 0.001 | 0.001 | 0.001 |

Table 8: Parameters for all evaluation metrics on each CV dataset.

## D.3 ADAPTION

In this section, we explain how we adapted the evaluation metrics in our framework to seamlessly work with 3D modalities. All metrics were adapted for point cloud (n,d) and volume (x,y,z) dimen-

sions besides classical image dimensions (w,h,c). We neglected the metrics which did not need any further adaption. All metrics leveraging threshold values expect normalized saliency maps on the observation level. Otherwise, thresholds have to be selected per observation.

**Pixel Flipping**    We compute the Area Under the Curve (AUC) to receive a single score. For point cloud data acts on the single coordinates.

**Region Perturbation**    We compute the AUC to receive a single score. Acts on a 3D kernel for volume data and single points for point cloud data. Compute the AUC to receive a single score.

**Insertion**    Use Gaussian noise for 3D data instead of Gaussian blur for images. Inserting single points for point cloud data and voxels for volume data.

**Deletion**    Deletes single points for point cloud data and voxels for volume data. Compute the AUC to receive a single score.

**Iterative Removal of Features (IROF)**    Compute the Area Over the Curve (AOC) to receive a single score. We use 3D Slic for volume segmentation and KMeans clustering with fixed $k = 16$ clusters for point cloud segmentation. $k = 16$ was determined by visual inspection. See exemplary visualization in Figure 5.

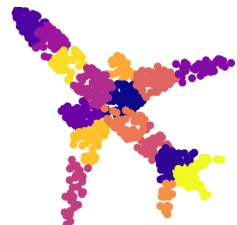

Figure 5: Example of KMeans clustering for point cloud data with k=16.

**Remove and Debias (ROAD)**    We use Gaussian noise for 3D modalities. Compute the AUC to receive a single score.

**Sufficiency**    Use the whole set of saliency maps for similarity comparison and not only the batch the metric is applied to (see Appendix J). For distance calculation between saliency maps, we use squared Euclidean distance for volume data and standardized Euclidean distance for image and point cloud data due to numerical instability.

**Continuity**    We implemented x-axis traversal for volume data along the x-axis with black padding in all dimensions and for the point cloud data by traversing all points along the x-axis position at $(n, d = 0)$ (see Figure 6). As removing points for point cloud data would change the input dimension of the object, we instead map them to the center (0,0,0). We did not observe any OOD behavior by implementing this solution. We use the Pearson Correlation Coefficient (PCC) between traversals to compute a single score.

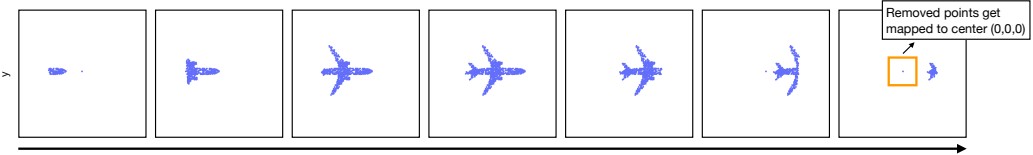

Figure 6: X-axis traversal of point clouds for continuity metric. We can not remove points as this would change the input dimensionality, thus we map them to the center (0,0,0), which is similar to black padding for image and volume data.

**Relative Representation Stability**    We use uniform noise ($U(0, 0.05)$) due to numerical stability as Gaussian noise could generate infinity values.

# E    RANKING COMPUTATION FLOW CHART

Figure 7 shows the transformation and aggregation steps from raw scores to final tables depending if we want to average across architectures or construct tables per architecture. In the calculation of the combinations, it must be taken into account that in the case of the Transformer architectures we have three more XAI methods (attention methods), and in the case of the point cloud modality, we have three fewer XAI methods (excluding CAM methods). In the case of the full ranking, we then have 7,560 combinations of CV datasets, architectures, XAI models, and evaluation metrics based on which we compute 50 scores for each combination, but always the same observations per dataset. If we average across architectures, we have to first normalize the 50 scores per architecture together, as the number of XAI methods differs between CNN and Transformer-based architectures. As we normalize across architectures we end up with 2,520 combinations but 150 scores per combination, which are in total again 378,000 scores. To receive the tables in the final step, we take the mean over the computer vision datasets and evaluation metrics per evaluation criteria, to receive one average rank per XAI method, evaluation criteria, modality, and depending if the ranking is full or across architectures, architecture. Values in the last aggregation step coincide with the number of scores per evaluation criteria, as each of the three criteria contains a different amount of metrics.

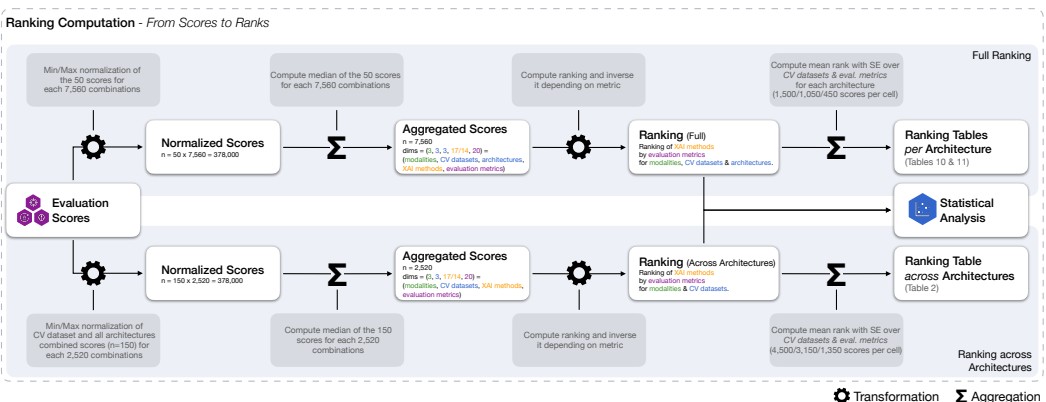

Figure 7: Transformation and aggregation steps from raw evaluation scores to final tables.

# F    ENSURING COMPARABILITY OF RESULTS

To ensure fair comparability between XAI methods over all possible experiment set-ups and aggregation levels, we take precautions about the XAI methods, evaluation metrics, and model architectures. Attribution measures the positive or negative contribution of an input feature (e.g. pixel) into the predicted output class of the model. On the contrary, CAM methods only compute positive attribution, and attention highlights all general (or absolute) important input features independent of the output class. However, in practice, attention is only valuable in interpretation if it also highlights features that are used for prediction. New methods such as LA filter the attention to only show such class-relevant attention, and their possible better performance to unfiltered attention can only be shown by evaluating it as positive attribution. Thus we consider only positive attribution for saliency map comparison (also suggested by Zhang et al. (2018)).

Further, we normalize the saliency maps on the observation level as some metrics have nominal thresholds or noise intensities which depend on the scale of saliency maps. As not all metrics compute single scores we have to convert all metrics computing sequences or array of sequences into single scores either via the AUC for Pixel Flipping, Region Perturbation, Selectivity and ROAD, AOC for IROF, or the PCC for SensitivityN and Continuity. All scores are normalized on the metric and dataset level. Score backpropagation-based metrics such as LRP (excluding the $0^+$-rule), DS or DLS, and the CAM methods expect non-negative activation outputs. Thus, we exchanged before the CNN model training all GeLU or leakyReLU activation functions with standard ReLU functions as they output negative values, biasing the XAI method. For the Transformer architectures, however, we keep all activation functions, as well as the skip connections and patchification, as they are central

to the architecture. Their potential effect on different attribution methods is part of the benchmark. For CAM methods on the Transformer architectures, we interpolate the reshaped *absolute* cls token, as saliency maps would otherwise often be empty (also recommended by Chefer et al. (2021)).

# G    METRIC-ANALYSIS FOR ALL CRITERIA

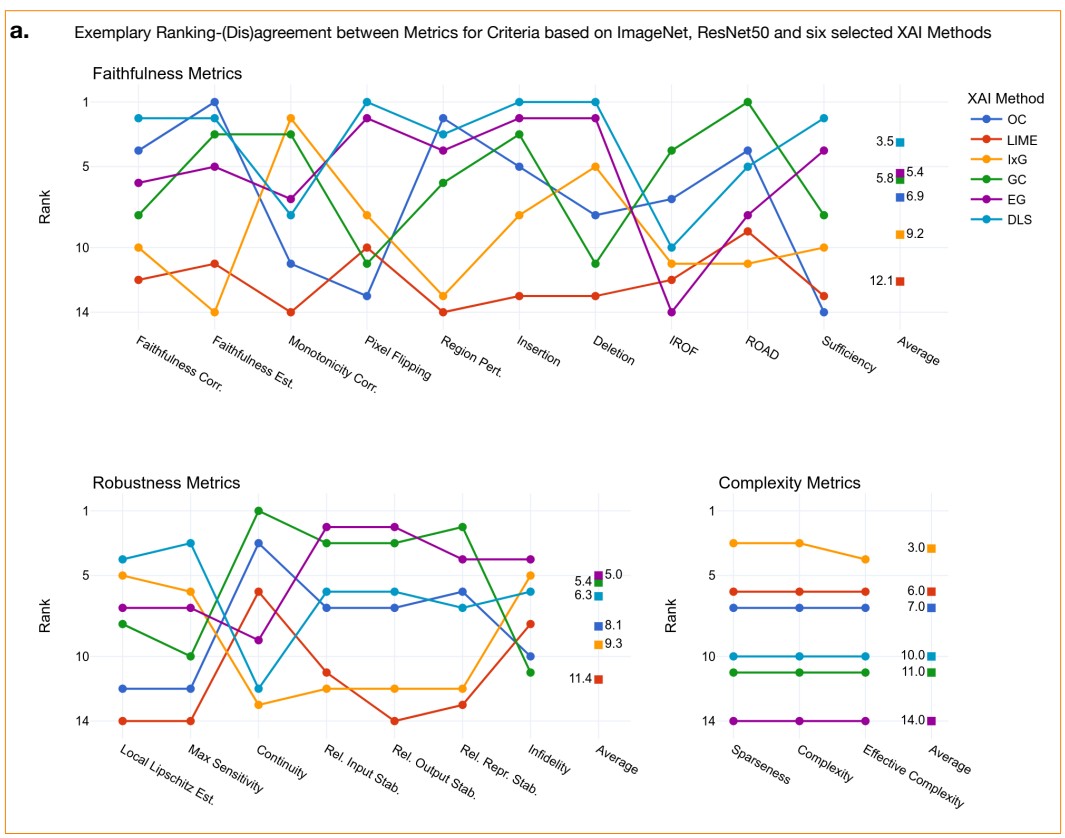

Figure 8: Distribution of ranks based on the different metrics for each criterion. This example shows only the ranking of six attribution methods for the IMN dataset and ResNet50 model. To the right of each plot, the average rank of each attribution method is shown. For certain attribution methods metrics tend to agree more (e.g. LIME and DLS on faithfulness) than others. However, inter-metric disagreement is important information for XAI evaluation.

Our study utilized a methodology of abstraction and aggregation into recognized criteria like faithfulness or robustness to simplify our analysis and strengthen the robustness of our findings. Such criteria are established in the field, with mean aggregation commonly used to synthesize scores and ranks Hesse et al. (2023); Bommer et al. (2023); Hedström et al. (2023a;b). For our aggregation, we incorporated standard metrics readily available in the popular computer vision XAI software packages Quantus (Hedström et al., 2023b) and Captum (Kokhlikyan et al., 2020), reflecting the field's most prevalent metrics. Nonetheless, questions persist regarding potential information loss due to aggregation and the robustness of these abstracted criteria. To address these concerns, we will initially provide a rationale for our methodologies, followed by an empirical analysis of our metrics for attribution methods within the image modality, with the aim of validating our assumptions and offering insights into metric behaviors.

To begin with, the lack of a reliable and general trend of aggregated ranks is not a sign of randomness, but a sign of robustness against outlier scores of individual metrics. A metric is always only a proxy for a criteria (faithfulness etc.) and is designed to be independent of the model architecture, utilized dataset, or other variables. Metrics are distinguished solely by their mathematical formula-

**a.** Mean Standard Deviation for Model Architecture and Utilized CV Dataset Dimensions

| Evaluation Criteria: | Model Architectures | | | Utilized CV Datasets | | |
|---|---|---|---|---|---|---|
| | ResNet50 | EffNetb0 | DeiT ViT | IMN | OCT | R45 |
| **Faithfulness** | 3.35 | 3.31 | 3.45 | 3.43 | 3.25 | 3.43 |
| **Robustness** | 3.19 | 3.2 | 2.96 | 3.25 | 3.09 | 3.01 |
| **Complexity** | 0.43 | 0.48 | 0.47 | 0.38 | 0.63 | 0.37 |
| *Weighted Average* | 2.86 | 2.85 | 2.83 | 2.91 | 2.80 | 2.82 |

Table 9: Mean standard deviation per model architectures and utilized CV datasets. The weighted average is based on the number of metrics per criterion to take into account the significance of the standard deviation of each criterion.

tions for computing these criteria. Thus, disagreement between metrics provides essential insights for XAI evaluation, reflecting the diversity of perspectives and implementations of the criteria. This important information is not even biasing the benchmark, as in the event of a lack of consensus among metrics, the average rank is expected to converge towards an uninformative mean rank, and only robust trends are detected. Statistically speaking, the sample of metrics is an unbiased estimator for the criteria. This is contrary to model performance metrics like e.g. F1 and accuracy, which measure different aspects of classification performance. Figure 8 demonstrates the ranking behavior of all metrics based on IMN, ResNet50, and six selected XAI methods. The line charts show the agreement and differences in ranking between metrics, with the average aggregated rank to the right. For e.g. faithfulness we observe as anticipated that XAI methods with high agreement between metrics such as LIME and DLS show more extreme ranks, while the other XAI methods with higher disagreement have average ranks converging against the mean rank of 7. Interestingly we observe almost no disagreement between the complexity metrics.

The inquiry emerges as to whether the variance seen in metric rankings is stochastic or systematic, and whether distinct patterns can be identified upon examining the variance through various dimensions. We computed the average standard deviation between metric rankings for each model and dataset dimension. The results are shown in Table 9. They show, that the average standard deviation is relatively stable within each evaluation criteria. For the model architectures, only the DeiT ViT standard deviation varies to a small extent for the faithfulness and robustness criteria compared to the two CNN models. For the utilized dataset dimensions, OCT complexity exhibits a higher SD than the other two datasets. However, the weighted average per column shows no major deviations. In general, we can conclude, that the variance between metric rankings is generally invariant to model architectures and dataset choices.

**a.** Proportion of smaller Metric Ranking Variance than Random Ranking Variance based on one-sided Levene Test for all Utilized Image Datasets and Model Architectures

| Evaluation Criteria: / Attribution Method: | OC | LI | KS | VG | IxG | GB | GC | SC | C+ | IG | EG | DL | DLS | LRP | *Average* |
|---|---|---|---|---|---|---|---|---|---|---|---|---|---|---|---|
| **Faithfulness** | 0.0 | 0.56 | 0.78 | 0.67 | 0.22 | 0.22 | 0.56 | 0.11 | 0.56 | 0.33 | 1.0 | 0.22 | 0.67 | 0.22 | 0.44 |
| **Robustness** | 0.33 | 0.56 | 1.0 | 0.56 | 0.0 | 0.56 | 0.44 | 0.22 | 0.33 | 0.11 | 0.33 | 0.0 | 0.89 | 0.33 | 0.4 |
| **Complexity** | 1.0 | 1.0 | 1.0 | 1.0 | 1.0 | 1.0 | 1.0 | 1.0 | 1.0 | 1.0 | 1.0 | 1.0 | 1.0 | 1.0 | 1.0 |
| *Weighted Average* | 0.27 | 0.63 | 0.89 | 0.68 | 0.26 | 0.46 | 0.58 | 0.28 | 0.55 | 0.35 | 0.77 | 0.26 | 0.8 | 0.38 | 0.51 |

> 0.7 ▇   < 0.3 ▇

Table 10: Proportion of accepted one-sided Levene-Tests for significant ($\alpha = 0.1$) smaller metric ranking variance than random ranking variance. Higher values show higher agreement between metrics of certain criteria. The weighted average at the bottom is computed by weighting the proportions per criteria by the number of metrics per criteria (i.e. 0.5, 0.35, and 0.15).

Nonetheless, such findings do not extend to the XAI methods, particularly attribution methods, an intended outcome that may be anticipated from the observations in Figure 8. To equitably and effectively quantify the differences in variance among attribution methods, we utilize a one-sided

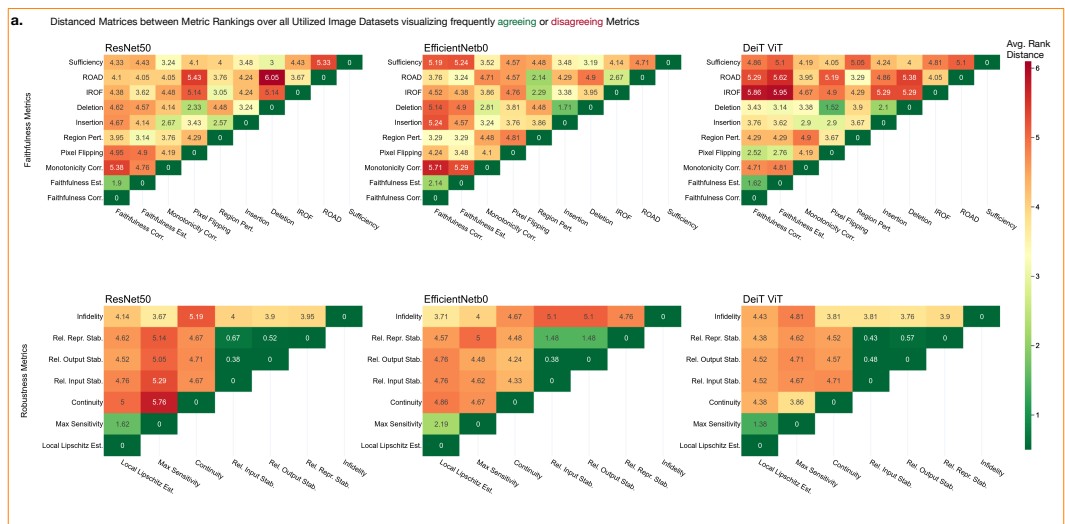

**Figure 9:** Average Euclidean ranking distance between metric pairs for model architectures and the faithfulness and robustness criteria. More often agreeing metric pairs in their rankings appear more green, and disagreeing pairs more red.

Levene's Test (Levene, 1960), testing if the rank-variance of a set of metrics is significant ($\alpha = 0.1$) lower than the variance of a total random rank distribution. The variance of a total random rank distribution can be analytically calculated through the variance of a discrete uniform $U_d(1, 14)$ distribution: $\sigma^2 = \frac{14^2 - 1}{12} = 16.25$. We compute this test for every combination of dataset, model architecture, attribution method, and evaluation criteria, and show the average acceptance rate of the test per attribution method and evaluation criteria in Table 10. A value converging against 1 indicates for all combinations statistically significant lower variance than random ranking, thus high agreement between metrics. Similar to Figure 8, we observe that there is almost no disagreement between complexity metrics. When computing the weighted average again at the bottom, we observe strong variations between the attribution methods. Specifically for KS, EG and DLS, in a large majority of cases, metrics agree, while for OC, IxG, SC, and DL only in about $\sim 27\%$ of the cases variance in metric ranking is lower than random ranking. It is important to mention that a high number of test rejections does not necessarily suggest a variance greater than that of random ranking. Our findings reveal that metrics tend to agree more for certain attribution methods, whereas others exhibit increased inter-rater variability. Investigating all underlying reasons for this discrepancy and why some attribution methods display greater inter-rater variability than others presents a compelling direction for future research.

In scenarios of high disagreement among metrics, potential biases could arise when considering only a limited subset of metrics, a concern we have raised in relation to other studies. Such scenarios further underscore our large-scale experimental design as it prevents undetected biases that could result from the selective use of individual metrics, intentional or accidental. Figure 9 shows the Euclidean distance, averaged over all attribution methods and datasets, between the ranking of two metrics for all model architectures and the faithfulness and robustness criteria. We observe that the general mean distance or disagreement between two metrics is around 4 ranks. We further observe around 8-10 outlier pairs for faithfulness and 2-5 for robustness, which have either substantially higher agreement ($\sim 2$ ranks) or higher disagreement ($\sim 5.5$ ranks). By first selecting a favorable metric and then pairing it with strongly similar ranking metrics based on this distance matrix (e.g. Faithfulness Estimate with Faithfulness Correlation), it is in theory possible to selectively pair similar ranking metrics to deliberately skew results towards a favorable outcome. However, this is only possible for a very small subset of metrics. Also the likelihood of any such "extremist subgroups" unduly influencing our large-scale study is small. The closest example of such subgroups would be the Relative Stability metrics for robustness.

We determined, that aggregation of metrics not only increases the robustness and large-scale studies employing several metrics but can give insight into metric disagreement, a valuable information. Further, our empirical metric analysis shows that aggregated ranks do indeed convey meaningful signals and are not significantly biased and hard to cheat in practise. Our inclusion of several model architectures and datasets also explicitly shows that they have no strong influence on the metric behavior (i.e. agreement and discrepancies). To ensure certainty beyond doubt, ranking independence from subgroup influence could also be ensured by using weighted averaging for the criteria by using the computed ranking distance matrices between the metrics. Other approaches including synthetic data or ground truth explanations have the advantage of computing metrics specifically independent of such dimensions. Hesse et al. (2023) propose the Background independence metric, which measures the extent to which a model's explanations are not overly sensitive to the entire image. It does this by focusing on the importance assigned to background objects that are deemed irrelevant to the model's decision. Further, Rao et al. (2022) showed the usefulness and accuracy of an explanation independent of the underlying model for graph neural networks on molecular data. First, they distilled the explanation(s) into an XAI-assisted fingerprint which they second used as the only input for a random forest model in molecular property prediction. The highly accurate predictions of the random forest classifier demonstrate the effectiveness of the XAI-assisted fingerprints.

# H ADDITIONAL RANKING TABLES

## H.1 FULL RANKING TABLE WITH STANDARD ERRORS

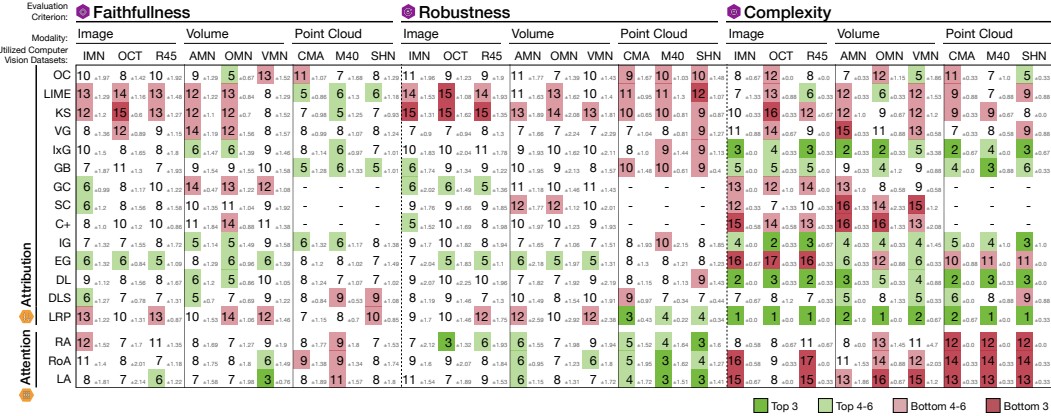

Table 11: Full ranking table for all XAI methods and CV datasets with standard error (SE).

## H.2 Ranking table CNNs only

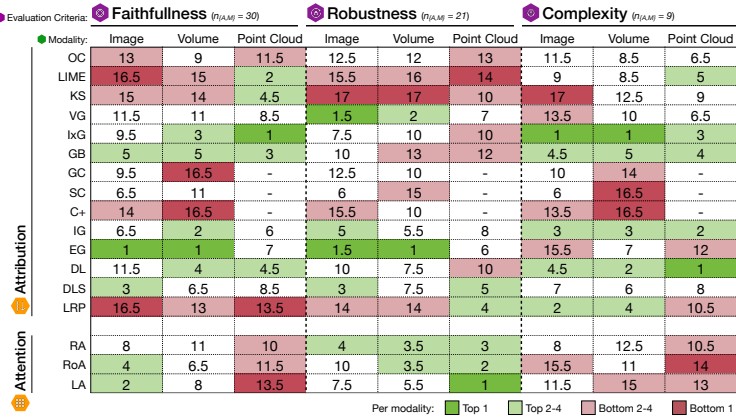

| Attribution | Faithfullness ($n_{(AM)} = 60$) | | | Robustness ($n_{(AM)} = 42$) | | | Complexity ($n_{(AM)} = 18$) | | |
|---|---|---|---|---|---|---|---|---|---|
| Modality: | Image | Volume | Point Cloud | Image | Volume | Point Cloud | Image | Volume | Point Cloud |
| OC | 8.5 | 5.5 | 9 | 11.5 | 5.5 | 4.5 | 9 | 10.5 | 7.5 |
| LIME | 14 | 9.5 | 2.5 | 14 | 12.5 | 11 | 6 | 8.5 | 7.5 |
| KS | 13 | 9.5 | 7 | 13 | 14 | 9 | 13.5 | 8.5 | 10 |
| VG | 11.5 | 11 | 9 | 6 | 2 | 2 | 10 | 13 | 9 |
| IxG | 10 | 2.5 | 5 | 11.5 | 9.5 | 7 | 3.5 | 1.5 | 3 |
| GB | 8.5 | 7 | 1 | 10 | 8 | 8 | 2 | 5 | 5.5 |
| GC | 3 | 14 | - | 2 | 11 | - | 12 | 7 | - |
| SC | 5 | 8 | - | 8 | 9.5 | - | 7.5 | 14 | - |
| C+ | 4 | 13 | - | 4.5 | 7 | - | 11 | 12 | - |
| IG | 6 | 1 | 5 | 8 | 3 | 4.5 | 5 | 4 | 4 |
| EG | 1 | 5.5 | 11 | 2 | 1 | 10 | 13.5 | 10.5 | 11 |
| DL | 7 | 2.5 | 2.5 | 8 | 4 | 4.5 | 1 | 3 | 2 |
| DLS | 2 | 4 | 9 | 4.5 | 5.5 | 1 | 7.5 | 6 | 5.5 |
| LRP | 11.5 | 12 | 5 | 2 | 12.5 | 4.5 | 3.5 | 1.5 | 1 |

Per modality: Top 1 — Top 2-4 — Bottom 2-4 — Bottom 1

Table 12: Ranking of the average rank over CNN architectures, datasets, and all evaluation metrics of the respective criteria, for each XAI method and modality (i.e. the rank of OC on image is based on $3 * 2 * 10 = 60$ ranks). Coloring coincides with top and bottom positions as no attention methods can be applied to CNN architectures.

## H.3 Ranking table Transformer only

| | | Faithfullness ($n_{(AM)} = 30$) | | | Robustness ($n_{(AM)} = 21$) | | | Complexity ($n_{(AM)} = 9$) | | |
|---|---|---|---|---|---|---|---|---|---|---|
| | Modality: | Image | Volume | Point Cloud | Image | Volume | Point Cloud | Image | Volume | Point Cloud |
| Attribution | OC | 13 | 9 | 11.5 | 12.5 | 12 | 13 | 11.5 | 8.5 | 6.5 |
| | LIME | 16.5 | 15 | 2 | 15.5 | 16 | 14 | 9 | 8.5 | 5 |
| | KS | 15 | 14 | 4.5 | 17 | 17 | 10 | 17 | 12.5 | 9 |
| | VG | 11.5 | 11 | 8.5 | 1.5 | 2 | 7 | 13.5 | 10 | 6.5 |
| | IxG | 9.5 | 3 | 1 | 7.5 | 10 | 10 | 1 | 1 | 3 |
| | GB | 5 | 5 | 3 | 10 | 13 | 12 | 4.5 | 5 | 4 |
| | GC | 9.5 | 16.5 | - | 12.5 | 10 | - | 10 | 14 | - |
| | SC | 6.5 | 11 | - | 6 | 15 | - | 6 | 16.5 | - |
| | C+ | 14 | 16.5 | - | 15.5 | 10 | - | 13.5 | 16.5 | - |
| | IG | 6.5 | 2 | 6 | 5 | 5.5 | 8 | 3 | 3 | 2 |
| | EG | 1 | 1 | 7 | 1.5 | 1 | 6 | 15.5 | 7 | 12 |
| | DL | 11.5 | 4 | 4.5 | 10 | 7.5 | 10 | 4.5 | 2 | 1 |
| | DLS | 3 | 6.5 | 8.5 | 3 | 7.5 | 5 | 7 | 6 | 8 |
| | LRP | 16.5 | 13 | 13.5 | 14 | 14 | 4 | 2 | 4 | 10.5 |
| Attention | RA | 8 | 11 | 10 | 4 | 3.5 | 3 | 8 | 12.5 | 10.5 |
| | RoA | 4 | 6.5 | 11.5 | 10 | 3.5 | 2 | 15.5 | 11 | 14 |
| | LA | 2 | 8 | 13.5 | 7.5 | 5.5 | 1 | 11.5 | 15 | 13 |

Per modality: Top 1 — Top 2-4 — Bottom 2-4 — Bottom 1

Table 13: Ranking of the average rank over Transformer architectures, datasets, and all evaluation metrics of the respective criteria, for each XAI method and modality (i.e. the rank of OC on image is based on $3 * 1 * 10 = 30$ ranks). Coloring coincides with top and bottom positions.

# I  DIFFERENCE IN FAITHFULNESS RANK FOR POINT CLOUD MODALITY

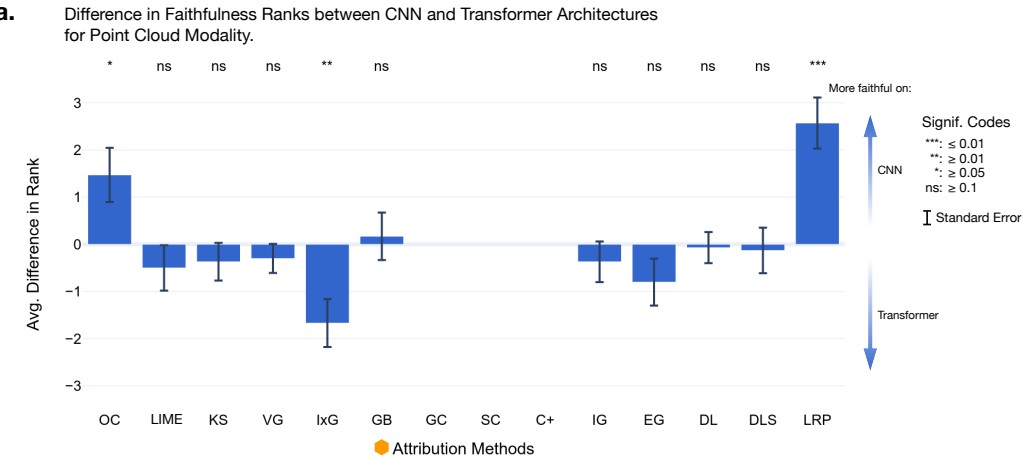

Figure 10: **a.** Average difference in rank when subtracting the average rank of CNN architectures from the rank of Transformer architectures over faithfulness metrics. T-test to validate if the difference significantly differs from 0.

# J  OPERATIONAL SHORTCOMINGS OF EVALUATION METRICS

While all metrics are theoretically very well founded, we observed for some metrics shortcomings in applications:

Sufficiency evaluates the likelihood that observations with the same saliency maps also share the same prediction label. In practice, this requires several saliency maps from observation with the same prediction label. While this works well on datasets with a small number of labels and balanced sampling, for datasets like IMN with 1000 labels, the probability is almost zero that at least 5-10 sampled observations in a set of sizes 50 or 100 have the same label (see Appendix K).

Sequence outputting metrics that alter the input space, such as Pixel Flipping, Region Perturbation, or ROAD, are only limited suitable for binary prediction tasks. When the input object is too noisy/perturbed to predict accurately, the probability for each class is 0.5 resulting in sequences converging against 0.5 and not 0. While the resulting AUC (or AOC in the case of Region Perturbation) can be compared between XAI methods within this task, between tasks the AUC would be biased as the area for the binary task would always be larger.

ROAD scores are arrays of binary sequences which are averaged to one sequence. The amount of noise has to be carefully tuned (also depending on the underlying model) as otherwise, all binary sequences in the array are only 0 or 1.

Local Lipschitz Estimate approximates the Lipschitz smoothness through several forward passes of a batch of observations. In application, this results in a large amount of RAM used (depending on modality) if the approximation should be stable. While the computation is relatively fast on a GPU, stable approximations exceed 40GB of VRAM by far and have to be partitioned. For the Transformer architectures, computation on the CPU for our amount of data was too slow to be feasible.

Effective complexity uses a nominal threshold value to determine attributed features. Even through normalization of the saliency maps, the threshold value can have a large effect on the results, differing between observations, and we would suggest tuning it per dataset.

IROF superpixel segmentation can result in very defined or binary structures such as in the AMN dataset in only two superpixels (object and background), ignoring finer structures.

As elaborated, all complexity metrics flatten the input object treating it as a vector and ignoring spatial dependencies.

## K Score distributions of evaluation metrics for all datasets

### K.1 Image modality

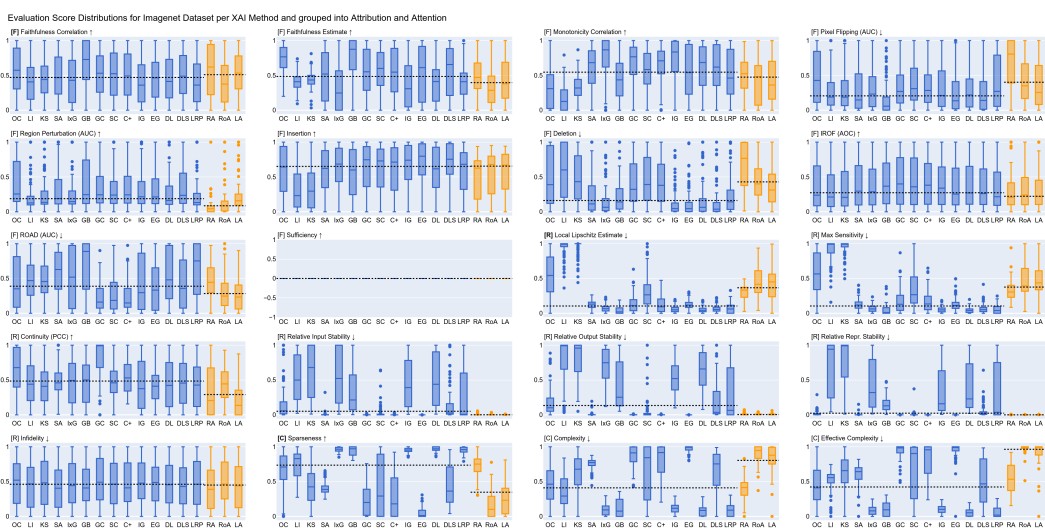

Figure 11: Score distributions of all evaluation metrics for each XAI method. Scores are normalized also for the Continuity PCC, as a negative correlation is worse than no correlation.

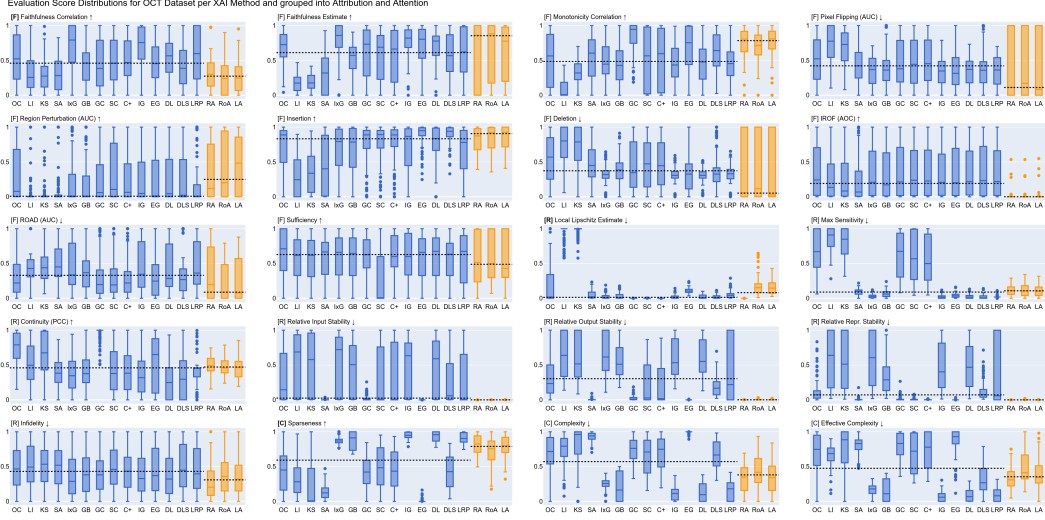

Figure 12: Score distributions of all evaluation metrics for each XAI method. Scores are normalized also for the Continuity PCC, as a negative correlation is worse than no correlation.

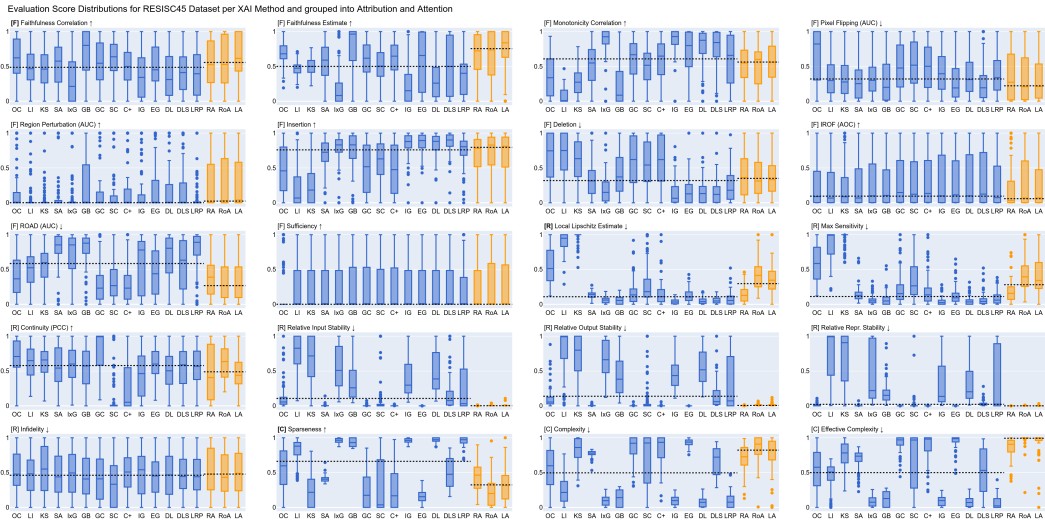

Figure 13: Score distributions of all evaluation metrics for each XAI method. Scores are normalized also for the Continuity PCC, as a negative correlation is worse than no correlation.

## K.2 VOLUME MODALITY

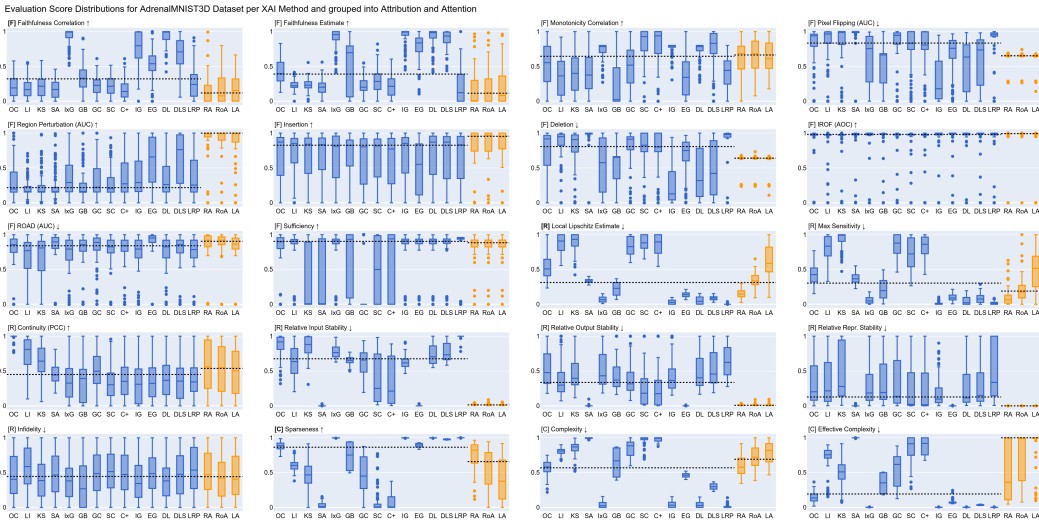

Figure 14: Score distributions of all evaluation metrics for each XAI method. Scores are normalized also for the Continuity PCC, as a negative correlation is worse than no correlation.

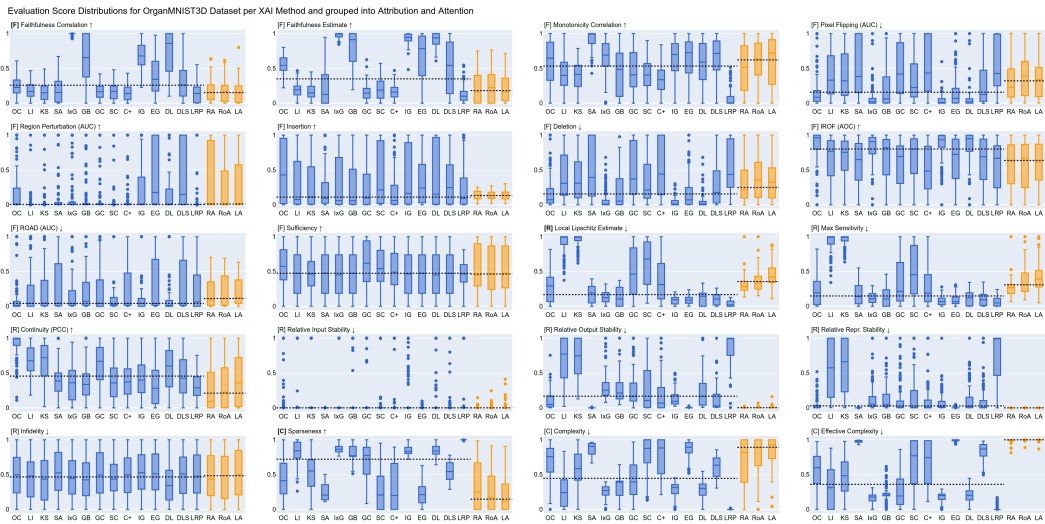

Figure 15: Score distributions of all evaluation metrics for each XAI method. Scores are normalized also for the Continuity PCC, as a negative correlation is worse than no correlation.

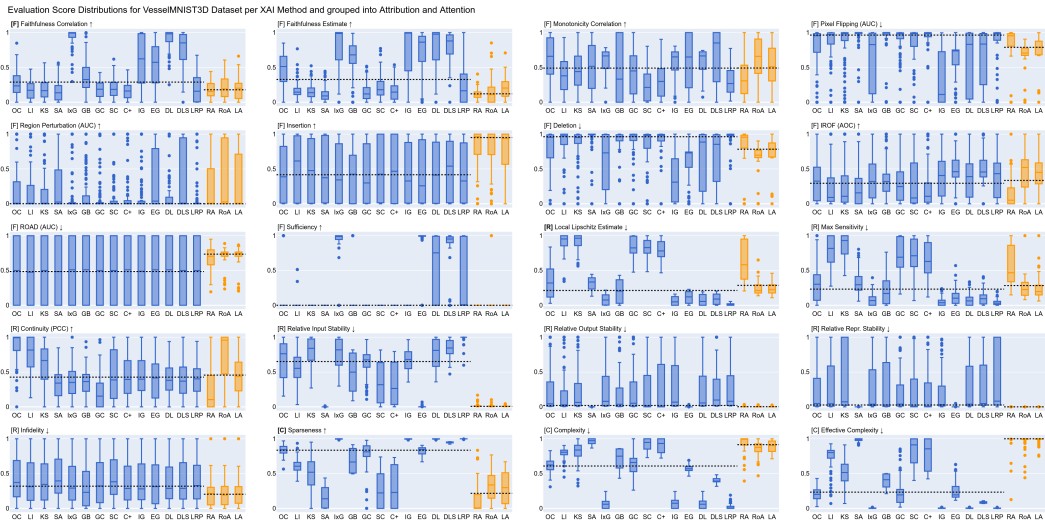

Figure 16: Score distributions of all evaluation metrics for each XAI method. Scores are normalized also for the Continuity PCC, as a negative correlation is worse than no correlation.

## K.3  POINT CLOUD MODALITY

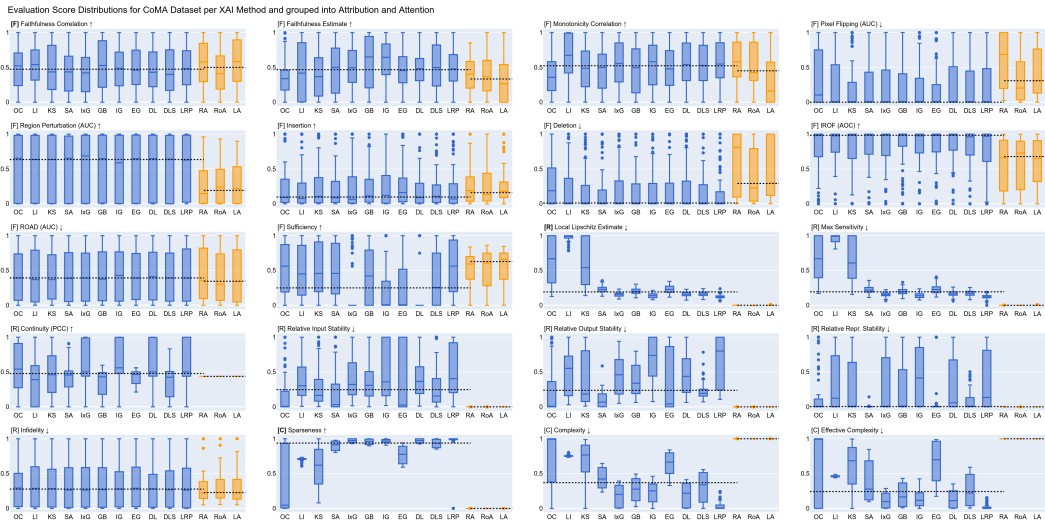

Figure 17: Score distributions of all evaluation metrics for each XAI method. Scores are normalized also for the Continuity PCC, as a negative correlation is worse than no correlation.

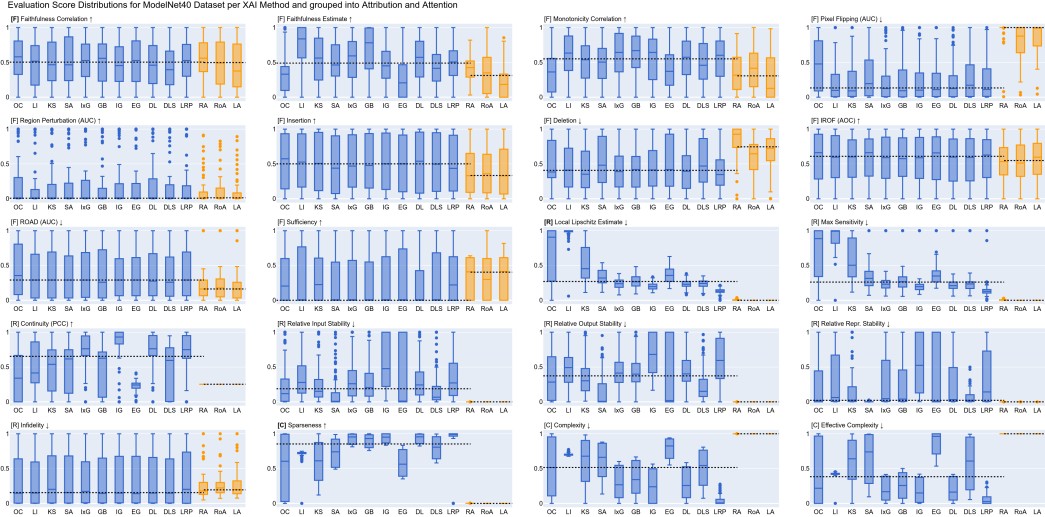

Figure 18: Score distributions of all evaluation metrics for each XAI method. Scores are normalized also for the Continuity PCC, as a negative correlation is worse than no correlation.

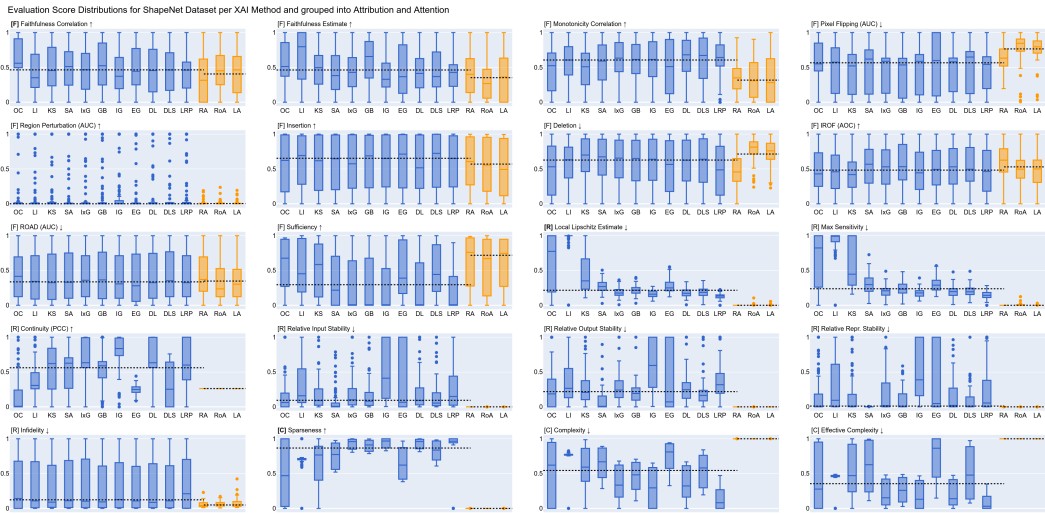

Figure 19: Score distributions of all evaluation metrics for each XAI method. Scores are normalized also for the Continuity PCC, as a negative correlation is worse than no correlation.

