# OpenReview forum: "LATEC — A benchmark for large-scale attribution & attention evaluation in computer vision"
_ICLR.cc/2024/Conference — Submitted to ICLR 2024_

### Official Review · Reviewer_aNQJ · 2023-10-24

**Soundness:** 3 good
**Presentation:** 2 fair
**Contribution:** 2 fair
**Rating:** 3
**Confidence:** 4

**Summary:**

The authors propose a framework for benchmarking attribution-based XAI methods, which combines many previously proposed metrics regarding faithfulness, robustness, and complexity of the attributions. Using this framework, the authors evaluate a wide range of different attribution methods on a variety of models and datasets, including CNNs and ViTs, as well as 2D, 3D, and point-cloud inputs.

The authors analyse and discuss their results and specifically aim to answer three questions they deem relevant in the context of attribution-based XAI methods, namely whether (1) attention-based attributions differ from 'classical' attribution methods, (2) whether the applicability of attribution methods depends on the modality, and (3) whether 'classical' methods still work for transformer models.

**Strengths:**

The following aspects that strengthen this submission:

**S1**: The experimental evaluation is very extensive, covering a wide range of attribution methods, models, and evaluation metrics on multiple datasets. To achieve this, the authors adapt existing approaches and models to point-cloud and 3D volume inputs when necessary.

**S2**: By making not only the code but various intermediate results publicly available (model weights, attribution maps, metric scores), the framework proposed by the authors allows for easily integrating additional evaluations, methods, and models. This is an important step to increase comparability and allow researchers to fairly test new XAI methods against existing approaches.

**S3**: The experimental details are clearly described and code for reproducing the results has been made available.

**Weaknesses:**

While the publicly available benchmark can surely prove useful to fellow researchers, I am hesitant to recommend the submission for acceptance for the following reasons.

**W1**: In order to deal with the large number of metrics across multiple datasets, modalities, and models, the authors heavily summarise the results into the broad categories of _faithfulness_, _robustness_, and _complexity_. While this seemingly allows to 'zoom out', I am concerned that this effectively hides much of the complexity of the design choices involved and makes it difficult to understand what one can really deduce from the results. E.g., as the authors note, the _complexity_, which is supposed to be a proxy for the subjective 'human interpretability' of the attributions does not seem to actually coincide with the perceived complexity. What is the value of this measure then? How do we know that the same is not true for faithfulness (see also W2)?

**W2**: For every subcategory, the authors essentially treat all metrics as being equally relevant and adequate for measuring, e.g. _faithfulness_. However, it is unclear to me whether it is meaningful to summarise these metrics into a single score, as each of them measure a different aspect of the models and define 'faithfulness' differently. This can also be observed in Fig. 9 in the appendix, as the different metrics for faithfulness do not seem to exhibit a reliable and general trend (compare, e.g., monotonicity and deletion). Further, they are sometimes based on assumptions that might not hold for a given model. E.g., models might be differently robust to pixel insertion or deletion and it is thus unclear whether these metrics measure aspects of the _underlying model_ or the _XAI method_. How stable are the findings when computing the ranks on different subsets of the metrics? If they are highly dependent on the chosen set of metrics, what do we really learn from the summary?

**W2b**: The authors missed highly relevant studies that also extensively study a wide range of attribution methods according to different criteria (e.g., Hesse et al., 2023, Rao et al. 2022). A comparative discussion to such works seems necessary, especially as those works explicitly try to avoid the pitfalls raised in W2 (i.e., disentangling model behaviour from the attribution methods).

**W3**: The implications and relevance of questions Q1-Q3 as well as their answers are not fully clear to me and I would appreciate if the authors could elaborate.

**W4 (minor)**: The manuscript is very dense and not easy to read and I encourage the authors to place a particular focus on making the writing more accessible. For example, using non-standard abbreviations for the various methods makes it tedious to read the main text, as one needs to constantly look up what a particular abbreviation stands for (e.g., LI for LIME or LA for 'LRP Attention', etc.). Further, it is difficult to understand a given table or figure from figure + caption alone, which makes the results much less accessible (this is heavily aggravated by the non-standard abbreviations, see e.g., Fig. 3c).

**W5 (minor)**: The relevance and implications of the discussion of Fig. 3a are unclear to me and the 'distinct clusters' that form seem to be an exaggeration of what is visible in the plots. Without the distinct colourings and the boxes drawn around some of the points, these points seem fairly uniformly distributed to me.

**Questions:**

Please see weaknesses. If the authors convince me that my concerns are unwarranted, I will consider raising my score.

---

> ### Author Response · Authors · 2023-11-17
>
> **Answer to W1 & W2:**\
> Thank you for this important feedback. We now provide an extensive metric-analysis in the paper (Appendix G) and modified the main text to reflect the increased awareness. In short, our analysis shows that aggregated ranks do indeed convey meaningful signals and are not significantly biased and hard to cheat in practice.
>
> We identified the main concern as the meaningfulness of our results given the potentially subjective selection and weighting in metric aggregation. Currently, studies use one or two metrics to reflect the criteria, leading to the displayed inconsistencies and even contradictions. Crucially, it is also common practice to aggregate over the (few) metrics (see e.g. Hesse et al. [1], brought up in your review, or Hedström et al. [2]). In contrast, our approach gets more robust insights by including a large number of adequate metrics per criterion. Adequacy is ensured by only selecting standard metrics in the field that are e.g. also implemented in the widely used Quantus library [2]. While individual metric-results are presented in Appendix K for assessing specific designs, they are too complex for readers; thus, we follow common practice in reporting aggregated ranks in the main paper for clarity.
>
> Concerning the meaningfulness of aggregated ranks, we argue that instead of measuring “different aspects” (like F1 and accuracy measure different aspects of classification performance), XAI metrics are rather nuanced proxy implementations aiming to measure the same specific aspect (e.g. faithfulness). Thus, aggregating over many of these proxies results in a robust rank invariant to individual implementations. Simultaneously, disagreement between several metrics provides essential insights for XAI evaluation, reflecting the diversity of implementations. Our new analysis shows that the aggregated ranks are indeed no meaningless mix, but in fact, convey meaningful signals based on the (dis)agreement of individual metrics. Further, we empirically see that variance in metric rankings is invariant to model architecture and dataset choice, and instead sensitive to XAI method selection as intended (see Appendix G for analysis using the Levene-Test).
>
> We agree that in scenarios of high disagreement among metrics, potential biases could arise when considering only a limited subset of metrics, a concern we have addressed in contrast to other studies via our large-scale setting. Our analysis suggests, that while it is in theory possible to selectively pair similar-ranking metrics to deliberately skew results, the actual risk of undue influence from such subgroups in our study is minimal.
>
> Our critique of complexity metrics, which can be arguably subjective, was methodological in nature and not based on metric disagreement, focusing on the unsuitability of their mathematical formulations for high-dimensional image data. This concern does not extend to faithfulness and robustness metrics, as these criteria are well-defined and more straightforward to express through mathematical mechanisms.
>
> **Answer to W2b:**\
> We thank you for highlighting these relevant studies. We've included [1] in the main manuscript. Moreover, in the added metric-analysis, we compare ourselves to their approach for metric aggregation and evaluation of the metric-sensitivity to the model. We would like to highlight that, unlike our real-world study, both approaches include explanation "ground truths". This makes it for example impossible for us to compute a metric like Background Independence from [1].
>
> **Answer to W3:**\
> Q1: Comprehending the differences between attention and attribution methods is crucial to determine whether new attention methods can be universally applied or if they present specific drawbacks or advantages over traditional methods.\
> Q2: While almost all XAI methods are designed for structured or image data, we don’t know if their applicability also transfers to other computer vision modalities, hindering their range of application.\
> Q3: Attribution methods are widely employed; however, it remains unclear whether their application to popular Transformer models parallels their use with CNNs. Thus ensuring their applicability with forthcoming DL architectures is crucial.
>
> We have updated the introduction of our work to express the significance of the three questions.
>
> **Answer to W4:**\
> We acknowledge that the abbreviations restrict readability. To address this issue, we have created a clean List of abbreviations on the first page of the appendix, which provides a quick and accessible reference for all abbreviations used in the manuscript.
>
> **Answer to W5:**\
> We agree that the term "distinct clusters" could be misleading and have toned down this statement to convey that similar operating methods tend to group together, which implies a non-random structure to the rankings.
>
> [2] Hedström et al. (2023). An explainable ai toolkit for responsible evaluation of neural network explanations and beyond.

---

> > ### Comment · Reviewer_aNQJ · 2023-11-21
> > **Answer to the authors and additional discussion**
> >
> > **Re W2b:** I appreciate that this concern has been addressed by the authors.
> >
> >
> > **Re W4:** I appreciate that the authors understand my concern and that a list of abbreviations has been included. I would still heavily recommend to use full names or common abbreviations where possible. E.g., why abbreviate LIME to LI in the first place? Why abbreviate GradCAM with GC and GradCAM++ with C+? There is very limited gain from the abbreviations introduced by the authors, but they heavily restrict readability. In this context, note also that DL is used to represent "Deep Learning" and "DeepLIFT".
> >
> >
> > **Re W5:** I appreciate that this concern has been addressed by the authors.
> >
> >
> > **W6 (reliability of the conclusions drawn for XAI methods):** The authors write on page 4: "XAI methods like LRP, DS, DLS, and CAM require non-negative activation outputs" without providing a reference for it. As far as I understand, this is an incorrect characterisation of the methods, and I believe all of them are able to handle negative activations; I would appreciate if the authors could explain why this should be the case and ideally provide references. Similarly, the claim that "[Architectural elements of the transformer] break the conservation rule for relevance backpropagation methods, (see [1] for a detailed analysis)" in [the response above](https://openreview.net/forum?id=tQYsKBTTaV&noteId=dXtg4kyfWv) is an inaccurate characterisation of the discussion in  [1] as well as of LRP as an XAI framework (see below).
> >
> > This brings me to a more general concern that I have, which is conceptually similar to W1: to make reliable statements about any given XAI method, it should be ensured that methods are used correctly and compared fairly. Otherwise, a benchmark as the one presented here might in fact *hurt* the research community by cementing certain views about XAI methods based on an inaccurate representation of a given method. This becomes most clear with LRP: specifically, LRP is not a monolith, but rather a general framework that subsumes many other approaches (e.g., depending on the rules chosen, it can be equivalent to Input x Gradient or the method presented by Chefer et al [1]). For an extension of LRP to transformers, i.e., one that defines attention-specific rules, see also Ali et al. (ICML, 2022, "XAI for Transformers").
> >
> > As mentioned in W1, my concern is that the heavy summarization for the metrics as well as the XAI methods makes it difficult to understand what we really learn. E.g., the authors say hyperparameters for the individual XAI methods were chosen based on a "qualitative evaluation" (page 3), which is highly subjective to begin with, but further does also not seem to include the LRP rules, despite the fact that (for CNNs at least) specific recommendations on which rules to use for which layers exist (see Montavon et al., 2019).  Given that the choice of hyperparameters (and rules for LRP) can drastically change the performance of most XAI methods, I am not convinced that the described procedure does justice to the methods that were evaluated.
> > This also directly implicates the findings regarding Q2: could the variations across modalities be due to the authors' adaptations and specific hyperparameter choices or is there a principled reason that some XAI methods are simply not compatible with other modalities?
> >
> > **W7 (novelty):** I also agree with the novelty concern raised by RKNb [see here](https://openreview.net/forum?id=tQYsKBTTaV&noteId=MKv9yJBgug). Specifically, apart from uploading intermediate results, I am not fully convinced that the proposed approach is significantly novel and different in its utility than the Quantus library itself (on which the authors also rely) or provides sufficiently novel and convincing insights; in fact, note that with 6 distinct evaluation categories, Quantus arguably allows for a broader understanding of a methods' behaviour. This concern is aggravated by my concerns regarding the insights we truly gain by simply aggregating the results of all metrics (W1+2) for a fixed choice of hyperparameters for each method which was chosen based on an unspecified qualitative inspection (W6).
> >
> > **W8 (minor):** Not all changes are integrated well and I would recommend that the authors carefully revise for readability. E.g.: The change on page 9 results in a sentence that seems to not make sense. Similarly, the newly added sentence on page 4 as well as the first sentence on page 2 (These questions...) are hard to understand.

---

> > > ### Author Response · Authors · 2023-11-22
> > >
> > > **Answer to W4:**\
> > > We agree that the abbreviations are not perfect, but since methods are mentioned very frequently, writing them all out would significantly extend the manuscript and hinder figure readability. To be consistent, we abbreviated all methods to two or max 3 letters. Nevertheless, we follow your suggestion and now do not abbreviate LIME, as we agree that in this case, the difference in length is insignificant. Notably, abbreviations do not seem to have negatively affected the readability for other reviewers. Thank you for pointing out the double assignment of DL, we revised it in the manuscript.
> > >
> > > **Answer to W6:**\
> > > Thank you for this comment. Indeed the first sentence is a missformulation. The requirement of non-negative activations is only true for LRP, the more general formulation in 3.3 is correct. We revised it accordingly. In the case of LRP, the relevance is computed through the previous neuron’s relevance times a weighted sum of the activation output times the LRP rule. If the activation output is negative, the term in the relevance sum will be negative, biasing the sum. Thus, the original LRP paper only considered ReLU activation functions [1, page 198]. A naive fix also used by [2] is only to consider the positive activations (e.g. $0^+$ Rule), but still biasing the relevance computation. For all Transformer architectures, we implemented as advised the $0^+$ Rule and for all CNN architectures, we followed the official recommendations of the original LRP paper and implemented the gamma rule for the lower layers and the epsilon rule for the upper and middle layers (as described in Appendix C.3).
> > >
> > > Further, [2] section 3.2, as well as Lemma 1, shows how in case of matrix multiplication in transformer architectures the conservation rule can not be preserved for LRP.
> > >
> > > We are aware that a benchmark may also have negative outcomes and results must always be placed in the context of the benchmark. However, in a real-world and large-scale benchmark, real-world and scaling circumstances have to be taken into account. Hyperparameter tuning for XAI methods was done qualitatively as every quantitative parameter evaluation would have biased the results and overfitted the methods to the metrics. Also, we argue that quantitative hyperparameter fitting for an XAI method is almost never done in practice. As described in Appendix C.2 all hyperparameters were first fitted on the dataset level, as the metrics, but we observed that the same parameters transferred between datasets. In the special case of LRP, we agree that the method is very versatile and can be changed through various rules. However, adapting the rules for every individual of the 9000 observations would be infeasible and thus not reflect application practice, and pose an unfair advantage over other methods. Thus we chose the specific recommended rules per layer type and position from the original LRP paper, which in our qualitative evaluation also resulted in convincing results.
> > >
> > > **Answer to W7:**\
> > > Quantus is a pure software library that provides a selection of metrics. Our work is significantly different from only a software implementation, as our full evaluation benchmark framework offers a comprehensive suite of tests and evaluations to assess the method’s overall performance and capabilities across several dimensions, which we in addition use for a large scale empirical study. Thus, these metrics only constitute a minor component of the overall benchmarking framework. Further, our work includes even additional metrics not included in Quantus (Insertion, Deletion, different form of Infidelity) and the metrics in Quantus only work with structured and image data. We extended all used metrics in the library to further work with volume and point cloud data. The localization and axiomatic evaluation criteria included in Quantus are not applicable in our benchmark, as the former requires ground truth bounding boxes for each observation and the latter is an XAI method-specific criterion, which cannot be compared across all XAI methods. We argue against defining  Randomization as its own criterion, as it only includes two very similar metrics which can be interpreted as a specific form of robustness against model parameter pertubations. While reviewer FKNb noted limited novelty regarding new metrics and more modalities, they recommended to accept the paper and acknowledged our argument that we first need comprehensive and foundational analysis before introducing additional complexity through more metrics and methods.
> > >
> > > **Answer to W8:**\
> > > We thank you for these suggestions and revised the manuscript for better readability.
> > >
> > > [1] Montavon, G. et al. (2019). Layer-wise relevance propagation: an overview.
> > > [2] Chefer, H. et al. (2021). Transformer interpretability beyond attention visualization.

---

### Official Review · Reviewer_dkvT · 2023-10-31

**Soundness:** 3 good
**Presentation:** 3 good
**Contribution:** 3 good
**Rating:** 6
**Confidence:** 2

**Summary:**

This paper proposes LATEC, a benchmark for large-scale attribution and attention evaluation in computer vision tasks. It evaluates 17 prominent XAI methods, with 20 different metrics. The dataset has been made publicly available.

**Strengths:**

* Overall I appreciate the effort of curating such a large-scale benchmark for XAI methods and metrics. I am not an expert in this particular field but I do believe it can be beneficial to the community.
* The presentation and description of the benchmark as well as the dataset is quite clear.
* The empirical study which aims to address the three pivotal questions makes a lot of sense.
* The discussion on insights and main takeaways are very clear and easy to follow.

**Weaknesses:**

* Particularly for faithfulness, from the dataset it is very much uniformly distributed. Does that mean this is not a meaningful metric to consider?
* Is there any interesting interplay between faithfulness, robustness and complexity?
* It is mentioned that theoretically biased XAI methods are less faithful on Transformers, but it is not clear why (from Sec. 3)? At least can you give some intuitions?
* Texts in some images, e.g. Fig 2 are a bit small and difficult to read.

**Questions:**

see above.

---

> ### Author Response · Authors · 2023-11-17
>
> **Answer to W1:**\
> The uniform distribution of faithfulness metrics across the highest level of aggregation indicates that this measure does not discriminate well between methods under these broad conditions. However, when we delve into specific groups (modalities, models, datasets, XAI methods), we observe variations that suggest faithfulness can indeed be a meaningful metric (see Table 2). Based on your appreciated feedback we also further explored and discussed this variation of faithfulness in specific conditions in the newly added Appendix section G, where we analyze metric behavior across different scenarios.
>
> **Answer to W2:**\
> We would argue that, theoretically, there can be a "trade-off" between faithfulness and complexity. A very faithful map can be very noisy, thus more complex, and very uncomplex maps such as CAM maps can be evaluated as quite unfaithful as they only highlight a general region of interest, not the exact pixels/features the model uses. Although a trade-off is noticeable for methods like LRP, GC, or EG, we advise against overinterpreting these findings in light of our reservations regarding the complexity metrics.
>
> **Answer to W3:**\
> Specific architectural choices in Transformer architectures, such as negative activation outputs, matrix multiplication in attention layers, or skip connections can, in theory, affect attribution methods. For example, this breaks the conservation rule for relevance backpropagation methods, (see [1] for a detailed analysis). In response to your valuable comment, we made the argument more prominent, especially in Section 3.
>
> **Answer to W4:**\
> We agree and increased the text size.
>
> We appreciate that there were no further significant wishes for improvement to the manuscript and hope to have addressed all questions to your satisfaction. Please let us know if there are any remaining concerns that could prevent you from increasing your score.
>
>
>
> [1] Chefer, H., Gur, S., & Wolf, L. (2021). Transformer interpretability beyond attention visualization. In Proceedings of the IEEE/CVF (pp. 782-791).

---

### Official Review · Reviewer_FKNb · 2023-11-03

**Soundness:** 4 excellent
**Presentation:** 4 excellent
**Contribution:** 3 good
**Rating:** 8
**Confidence:** 4

**Summary:**

The paper proposes a benchmark for XAI methods LATEC.

- *Methods to benchmarks:* This benchmark includes 17 different methods, this include both traditional (i.e saliency/gradient-based) methods trained on CNNs and Transformers and attention-based methods trained on transformers.

- *Datasets:* The benchmark considers 3 computer vision data modalities Images, volume(3D data) and point cloud. For each modality 3 different datasets were considered.

- *Benchmarking Metrics:* The paper investigates 3 aspects of XAI methods faithfulness, robustness and complexity. While the paper does not introduce any metrics itself it investigates the performance of different XAI methods on 20 previously proposed metrics and groups them into the 3 aspects.

 Overall the LATEC consists of 7,560 different combinations that were tested, after running this benchmark the paper summarizes the takeaways as follows:
- No XAI method ranks consistently high on all evaluation criteria.
- The rankings of XAI methods generalize well over datasets from the same modality.
- The complexity metrics proposed for CV tasks does not always have to match the perception of low complexity.

The paper then uses LATEC to answer 3 open XAI questions:

 - How does the performance of attention versus attribution methods differ in practice? They found a large difference in complexity (attention methods are more complex)  and a smaller difference in robustness  (attribution methods are more robust), while the difference in faithfulness is substantially insignificant. However, the paper does mention that the complexity methods are generally debatable since they favor methods that attribute to the smallest set of single pixels.

- Does the efficacy of XAI methods vary across different computer vision modalities? The performance varies across different modalities.

- With the ascendency of Transformer architectures, is there a potential misalignment with established attribution-based XAI methods?  The benchmark showed that the faithfulness of different attribution methods highly fluctuates for transformers and that attribution based methods are not necessarily a wrong choice when using a transformer architecture.

**Strengths:**

The main advantage of this benchmark is:
- (a) It considers different neural architectures.
- (b) It considers different data modalities.
- (c) It considers both traditional XAI methods and attention-based methods for transformers.
- (d) It considers different aspects in XAI that researchers or users might care about (i.e faithfulness, robustness and complexity).
- (e) Most of the popular XAI metrics were included.
- (f) It's very comprehensive overall, 7,560 different combinations were tested.
-(g) The code is structured in a way that it is very easy to add to either a new method or a new metric, which can make it an excellent resource for open-source collaborations in the future.


The paper is well-written and easy to follow. The takeaways from the experiments are clearly stated.

**Weaknesses:**

Novelty is limited: No new datasets or evaluation metrics were introduced in this benchmark.

Although different data modalities were considered, this benchmark only applies to computer vision tasks, the performance of different   XAI methods on other data types like tabular, time series, and language was not investigated.

**Questions:**

- Do you think the ranking of methods would have varied if synthetic baselines with ground-truth were included?

---

> ### Author Response · Authors · 2023-11-17
>
> **Answer to W1:**\
> We acknowledge the reviewer's perspective on the novelty of our benchmark; however, we posit that the proliferation of datasets, metrics, and methods in the field necessitates a comprehensive and foundational analysis before introducing additional complexity. Our benchmark demonstrates that rankings are robust across datasets, indicating that the diversity of datasets is not a constraint on the insights gained. Moreover, by making all intermediate steps of our evaluation process publicly available, we have laid the groundwork for the easy integration of new metrics and methods into our framework. This extensibility is a significant contribution to our work and one which we will emphasize further in the conclusion to underscore the benchmark's potential as a foundational platform for future explorations in the field.
>
> **Answer to W2:**\
> We agree with the significance of multi-modality in XAI research. However, we believe that expanding beyond the three modalities covered in this paper would have exceeded its intended scope. We fully support and anticipate future extensions of our benchmark to encompass additional modalities, and we have already earmarked this as a direction for our future research endeavors.
>
> **Answer to Q1:**\
> In our opinion, this would have heavily depended on the definition and implementation of ground truth and the synthetic baseline. Arras et al. [1] stated: “In many cases there probably exists no perfect Ground Truth” and there is a trade-off between realism and evaluation precision for synthetic evaluation data. However, the inclusion of a ground truth on synthetic data allows for much more detailed metrics regarding faithfulness and robustness (see [2]) as well as simple accuracy metrics for localization (see [3]). While the implications of these metric-results for real-world contexts such as ours remain unclear, exploring the transferability of results from artificial to real-world settings presents an intriguing avenue for future research in metric and benchmark evaluation.
>
> We appreciate your indication that there are no significant remaining requests for enhancements to our manuscript and thank you for the insightful question and the interest to already extending the benchmark. Please let us know if there are any remaining concerns that could prevent you from increasing your score.
>
> [1] Arras, L., Osman, A., & Samek, W. (2022). CLEVR-XAI: A benchmark dataset for the ground truth evaluation of neural network explanations. Information Fusion, 81, 14-40.\
> [2] Hesse, R., Schaub-Meyer, S., & Roth, S. (2023). FunnyBirds: A Synthetic Vision Dataset for a Part-Based Analysis of Explainable AI Methods. In Proceedings of the IEEE/CVF (pp. 3981-3991).\
> [3] Zhang, J., Bargal, S. A. et al. (2018). Top-down neural attention by excitation backprop. International Journal of Computer Vision, 126(10), 1084-1102.

---

> > ### Comment · Reviewer_FKNb · 2023-11-20
> > **Thank you for your response**
> >
> > I would like to thank the authors for their response.
> > My score remains as is.

---

### Official Review · Reviewer_Kney · 2023-11-03

**Soundness:** 2 fair
**Presentation:** 1 poor
**Contribution:** 3 good
**Rating:** 5
**Confidence:** 3

**Summary:**

Explainable AI (XAI) is an important and rapidly growing research area that aims to propose XAI methods for a better understanding of complex machine learning decisions. This paper performs a comprehensive evaluation of existing XAI methods to analyze their advantages and disadvantages in three aspects, including faithfulness, robustness, and complexity. Specifically, the proposed LATEC is a large-scale benchmark that evaluates 17 XAI methods using 20 distinct metrics. Furthermore, it extends the evaluation from 2D images to 3D point clouds. As a result, it incorporates vital elements like varied architectures and diverse input types, resulting in 7,560 examined combinations. The code, models, and data are publicly available.

**Strengths:**

1. The proposed LATEC performs an extensive evaluation of 17 XAI methods using 20 distinct metrics, which incorporates 7,560 examined combinations. The evaluation results are solid.
2. LATEC proposes solutions to extend the evaluation of XAI methods from 2D images to 3D point clouds, leading to more comprehensive benchmark results.
3. The code, models, and data are available to evaluate customized XAI methods or metrics.

**Weaknesses:**

1. The presentation is hard to read and understand. It is an experimental report rather than a well-organized research paper.
2. As a benchmark and analysis work, The takeaways from this paper are not insightful but rather just some straightforward observations. After reading this paper, I am not able to gain good insights about the good way to evaluate XAI methods.

Overall, this paper makes solid experiments. However, it fails to reveal convincing explanations and insightful comments. This paper is an experimental report rather than a well-prepared paper.

**Questions:**

See weakness

---

> ### Author Response · Authors · 2023-11-17
>
> **Answer to W1:**\
> We thank you for your critique regarding the presentation of our manuscript. We would like to point out that our submission aligns with the structure and style that is customary for the benchmark and dataset track of this conference, which should not be confused with the structure of a classical method proposal. Exemplary, we selected five previously accepted and outstanding ranked ICLR benchmark and dataset papers, that have similar structure: Jaeger et al. (Notable Top 5%) [1],  Galil et al. (Notable Top 25%) [2], Vaze et al. (Oral) [3], Zong et al. (Notable Top 25%) [4] and Galil et al. [5]. We followed these successful ICLR papers in their structure:
> - Formulating/Motivating Research Question
> - Benchmark Design
> - Experiment Evaluation
> - Insights and Take-aways.
>
> Notably all other reviewers specifically acknowledged that our paper is “well thought out presentation” (“6LcT”), “well-written and easy to follow” (“FKNb”), “presentation and description of the benchmark [...] is quite clear” (“dkvT”) and “experimental details are clearly described” (“aNQJ”). We would greatly appreciate it if you could specify the sections that you found to be lacking in organization so that we may enhance those areas.
>
> **Answer to W2:**\
> We would like to clarify that, while our evaluative strategies may not differ conceptually from existing ones (and were not intended to), the scale of our analysis is unprecedented and the dimensionality is unique. The resulting insights regard the application of XAI methods, not their evaluation, and exemplary show how relevant knowledge can be derived from our benchmarking.
>
> Specifically, our benchmark offers a practical tool for practitioners to test various XAI methods across different metrics suited to their specific problem configurations. For researchers developing new XAI methods or metrics, our recommendations for testing across diverse datasets, modalities, and model architectures ensure robust and meaningful insights. Only because we followed these recommendations, we resolved prior inconsistencies in method evaluations (e.g. ignoring attention methods and modalities besides images) and showcased how benchmarking according to these recommendations is highly relevant. We have amended the manuscript to better highlight these contributions and ensure the insights gained are clear and actionable.
>
> We thank you for pointing out the lack of clarity in this context and believe that our respective changes will avoid future confusion. As we believe to have resolved your comments, please let us know if there are any remaining concerns that could prevent you from recommending acceptance.
>
> [1] Jaeger, P. F. et al. (2022). A call to reflect on evaluation practices for failure detection in image classification.\
> [2] Galil, I.  et al. (2023). A framework for benchmarking class-out-of-distribution detection and its application to ImageNet.\
> [3] Wiles, O. et al (2021). A fine-grained analysis on distribution shift.\
> [4] Zong, Y. et al.  (2022). MEDFAIR: Benchmarking fairness for medical imaging.\
> [5] Galil, I. et al. (2023). What can we learn from the selective prediction and uncertainty estimation performance of 523 imagenet classifiers.

---

### Official Review · Reviewer_6LcT · 2023-11-03

**Soundness:** 4 excellent
**Presentation:** 4 excellent
**Contribution:** 4 excellent
**Rating:** 6
**Confidence:** 3

**Summary:**

The authors developed a large-scale benchmark that critically evaluates 17 XAI methods using 20 distinct metrics, using three guiding questions of current interest. The authors perform non-trivial statistical analyses to compare different methods along 3 evaluation criteria (faithfulness, robustness, and complexity), with special attention to the specifics of each method/dataset, as well as the interplay between attribution and attention (as in Transformer models). The authors go the extra step by making all their data available, e.g., model weights and saliency maps, to further facilitate future work.

**Strengths:**

- Well thought out presentation, detailing all steps taken and analyses applied, with high quality figures.
- Excellent reproducibility.
- Evaluates on point cloud and image volume inputs, beyond traditional evaluations on 2D images.
- Clear summary of main findings and takeaways.

**Weaknesses:**

Nothing stands out beyond the few points raised below.

**Questions:**

**Presentation:**
- Section 1:
    - The citations on the first few sentences aren't directly tied to the citing text. Perhaps better references can be used, or the text can be modified slightly to better link to those cited article. Please clarify whether those articles provide supporting evidence, focused on subproblems or application domains, or whether their authors simply echoed similar opinions.
    - Follow-up: it seems this discussion at the beginning is focused on saliency maps specifically. If so, please clarify the scope on the onset to be focused on XAI for computer vision models, with saliency methods as the primary approach being discussed. General statements about state-of-the-art in XAI would require an equally general selection of citations.
    - Indeed, the authors seem to repeatedly use "XAI research" to solely mean works on computer vision.
    - Please also clarify why saliency methods were chosen as the primary approach for this study, with a forward reference to a literature review highlighting other approaches.
- Section 2:
    - Please include a reference to Fig.1 in the main text.
    - S2.1: it would help to include a brief description of each XAI method, e.g., in an appendix, together with references to recent surveys. (perhaps in a form similar to Appendix B.2)
    - Figure 3/4: it seems the legend shows significance indicators that don't actually appear in the figures.

**Nitpicking:**
- Page 3: without adaptions -> adaptation? (I found 7 or so other occurrences)
- S2.2: Due to the LATEC dataset -> Thanks to? Using?
- S3.1: Transformer architecture inherit attention -> inherent?
- Page 6: commendable more robustness

---

> ### Author Response · Authors · 2023-11-17
>
> **Answer to Q1 (Section 1):**\
> Thank you for your valuable feedback regarding the citations in Section 1 of our manuscript. Upon reviewing the section, we concur that the initial references were not optimally aligned with the corresponding text. To address this, we have carefully revised the text to more accurately reflect and connect with the cited articles. While most citations reflect the definition of a term or method (e.g. for faithfulness, robustness, or complexity), others give supporting evidence (e.g. (Adebayo et al., 2018) or (Wang et al. (2019); Rosenfeld (2021))), or are examples for applications (e.g. Doshi-Velez& Kim (2017); Ribeiro et al. (2016); Shrikumar et al. (2017)).
>
> **Answer to Q2 (Section 1):**\
> We agree with your observation that the term "XAI methods" was initially defined too broadly. Accordingly, we have refined the manuscript to specify our focus on saliency map methods right from the beginning.
>
> **Answer to Q3 (Section 1):**\
> We appreciate the opportunity to clarify the scope of our research. The shortcomings identified in our work, though applicable across various domains of XAI research, are indeed addressed within the confines of computer vision modalities in this paper. In light of this, we have revised the manuscript to clearly define our focus on computer vision within the broader context of XAI research. We have made the necessary adjustments in the manuscript to ensure that our narrative accurately represents the scope and boundaries of our work.
>
> **Answer to Q4 (Section 1):**\
> In our manuscript, we posit that saliency maps represent the most prevalent post-hoc XAI methods in computer vision. These maps, while primarily illuminating "what" aspects of the data are influential in a model's decision-making process rather than "why," continue to form the foundational basis of most XAI analyses [1,2,3]. We underscore the criticality of trustworthy saliency maps at this stage, as mistakes can have critical consequences in subsequent analysis, leading to a misdirected emphasis on irrelevant input features. Moreover, we highlight the evolving sophistication of saliency maps, as advancements in attention and concept methods [4] show promise for the future of saliency maps. It is also noteworthy that post-hoc methods such as saliency maps offer ease of integration with existing model training and evaluation workflows, a practical advantage over ante-hoc XAI methods.
>
> In response to the feedback, we have adapted the manuscript to provide a more comprehensive overview of the XAI field, emphasizing the significance of saliency methods within this domain. We have included references to two exemplary approaches, counterfactual examples, and concept testing, and cited an extensive literature review to offer readers a full perspective of the field.
>
> **Answer to Q5 (Section 2):**\
> Thank you for pointing out the oversight. We have now included a direct reference to Figure 1 in Section 2.
>
> **Answer to Q6 (Section 2):**\
> We agree that this would be beneficial for readers who are new to the field, or just miss the knowledge about a specific method. In accordance with your advice, we have included a new section in the appendix that offers concise summaries of the XAI methods discussed, akin to the structure presented in Appendix B.2. We trust that this addition will enrich the informational value of our work for all readers.
>
> **Answer to Q7 (Section 2):**\
> Thank you for bringing this to our attention. The significance indicators are indeed present above the figures but are rendered at a size that may compromise their visibility. We acknowledge this oversight and have since adjusted the size of the significance indicators to ensure they are clearly visible within the figures.
>
> **Answer to Q8 (Nitpicking):**\
> We thank you for the comments and revised the manuscript accordingly.
>
> We appreciate the thoughtful questions you raised, as they have significantly contributed to the improvement in the clarity and explicitness of our manuscript. Please let us know if there are any remaining concerns that could prevent you from increasing your score.
>
>
> [1] Hauser, K. et al. (2022). Explainable artificial intelligence in skin cancer recognition: A systematic review. European Journal of Cancer, 167.\
> [2] Gevaert, C. M. (2022). Explainable AI for earth observation: A review including societal and regulatory perspectives. International Journal of Applied Earth Observation and Geoinformation, 112.\
> [3] Van der Velden, B. H. et al. (2022). Explainable artificial intelligence (XAI) in deep learning-based medical image analysis. Medical Image Analysis, 79.\
> [4] Achtibat, R. et al. (2023). From attribution maps to human-understandable explanations through Concept Relevance Propagation. Nature Machine Intelligence, 5(9).

---

> > ### Comment · Reviewer_6LcT · 2023-11-21
> > **Acknowledgement**
> >
> > Thank you for addressing my comments.
> >
> > I will be revising my score based on the discussion with other reviewer.

---

### Author Response · Authors · 2023-11-17
**Answer to all Authors**

We sincerely thank all five reviewers for their valuable comments. While the reviewers generally agreed on the added value of our work (“Excellent reproducibility”, “paper is well-written and easy to follow”, “important step to increase comparability”, “appreciate the effort of curating such a large-scale benchmark”), concerns were raised regarding the aggregation of metrics (“aNQJ”) and the organization of the paper (“Kney”).

We have thoroughly revised our manuscript to clarify the respective parts and resolve these concerns. We address all comments point-by-point below for each reviewer.

Changes in the revised manuscript are highlighted in orange.

---

### Author Response · Authors · 2023-11-23

We would like to respectfully point out, that despite the instructions for reviewers to acknowledge authors’ responses, it seems Reviewers “Kney” and “dkvT” have not recognized our rebuttal nor the significant updates made to the manuscript, which directly address and resolve their initial concerns.

---

### Meta-Review · Area_Chair_kV6X · 2023-12-06

**Metareview:**

The authors perform a large-scale evaluation of 17 XAI methods in computer vision across different modalities (image, volume, point cloud). This benchmark is more comprehensive than existing studies in this area, and the authors include evaluations on 20 metrics proposed across different works, grouping them into measurements of faithfulness, robustness, and complexity. Reviewers agree on the impressive scale of the benchmark; however, several of the reviewers had a discussion on whether this paper provides sufficient depth of analysis beyond its scale (and whether the XAI community would benefit from this work). Ultimately, I agree with concerns raised by reviewers that in it's current form, the paper may not provide significant insights on evaluations of XAI methods (other than the numbers/observations in the experiments), and the paper also lacks a detailed discussion on the choice of metrics and their grouping (instead there is a list in Appendix D). I do think this line of work is important, and I encourage the authors to revise their work based on the detailed feedback from reviewers - with a greater depth of analysis (and further analysis on the reliability of these metrics, perhaps through a user study), this paper will be valuable for the XAI and CV communities.

**Justification For Why Not Higher Score:**

While all reviewers agree on the scale of the evaluation, it is unclear whether the observations in this work provides significant insight for the XAI community. Furthermore, this is a benchmarking paper where the metrics are crucial, and in it's current form, the paper lacks a detailed discussion and justification of the choice of metrics used / grouping of metrics, as well as the reliability of the metric choices. For example, the authors have noted limitations of some existing metrics in the paper - one potential way to improve this work is to distill these observations into a new metric in order to move the field forward.

**Justification For Why Not Lower Score:**

N/A

---

### Decision · Program_Chairs · 2024-01-16

Reject